# Modeling the contribution of leads to sea spray aerosol in the high Arctic

Rémy Lapere[1,2], Louis Marelle[2], Pierre Rampal[1], Laurent Brodeau[1,4], Christian Melsheimer[3], Gunnar Spreen[3], and Jennie L. Thomas[1]

[1]Univ. Grenoble Alpes, CNRS, INRAE, IRD, Grenoble INP, IGE, 38000 Grenoble, France
[2]LATMOS/IPSL, Sorbonne Université, UVSQ, CNRS, Paris, France
[3]Institute of Environmental Physics (IUP), University of Bremen, Bremen, Germany
[4]DATLAS, Grenoble, France

**Correspondence:** Rémy Lapere (remy.lapere@latmos.ipsl.fr) and Jennie L. Thomas (jennie.thomas@univ-grenoble-alpes.fr)

**Abstract.** Elongated open water areas in sea ice (leads) release sea spray particles to the atmosphere. However, there is limited knowledge on the amount, properties and drivers of sea spray emitted from leads, and no existing parameterization of this process is available for use in models. In this work, we use measurements of aerosol fluxes from Nilsson et al. (2001) to produce an estimate of the location, timing and amount of sea spray emissions from leads at the scale of the Arctic Ocean for one year. Lead fractions are derived using sea ice data sets from numerical models and satellite detection. The proposed parameterization estimates that leads account for 0.3%–9.8% of the annual sea salt aerosol number emissions in the Arctic Ocean regions where sea ice concentration is greater than 80%. Assuming similar size distribution as emissions from the open ocean, leads account for 30%–85% of mass emissions in sea ice regions. The total annual mass of sea salt emitted from leads, 0.1–2.1 Tg yr$^{-1}$, is comparable to the mass of sea salt aerosol transported above sea ice from the open ocean, according to the MERRA-2 reanalysis. In addition to providing the first estimates of possible upper and lower bounds of sea spray emissions from leads, the conceptual model developed in this work is implemented and tested in the regional atmospheric chemistry model WRF-Chem. Given the estimates obtained in this work, the impact of sea spray from leads on Arctic clouds and radiative budget needs to be further explored.

# 1 Introduction

Aerosols resulting from sea spray make up most of the aerosol mass over the polar regions (Sand et al., 2017) and are a critical driver of polar climate (Struthers et al., 2011; Lapere et al., 2023). *Sea spray* is the mix of *sea salt* and *organics* co-emitted together from oceanic sources. These aerosols impact clouds, precipitation and climate, as they can act as Cloud Condensation Nuclei (CCN) or Ice Nucleating Particles (INP) depending on their composition (i.e. fraction of inorganic salts and organic matter) and processing in the atmosphere (Quinn et al., 2017; Fossum et al., 2018; Wilson et al., 2015; DeMott et al., 2016). Sea spray particles also contribute to the aerosol direct radiative effect by scattering incoming solar shortwave radiation (Takemura et al., 2002; Satheesh and Lubin, 2003). In addition, sea spray aerosols can change the climate impact of aerosols of other origin through mixing, such as nitrate (Chen et al., 2020) and sulfate (Fossum et al., 2020), by regulating their droplet activation. Similarly, changes in sea spray aerosols can impact the condensation sink in the Arctic, which in turn affects new particle formation and therefore the CCN populations (Browse et al., 2014), although results vary for different models (Gilgen et al., 2018).

Sea spray emissions in the mid-latitude oceans are driven by wind action that generates whitecaps, which release aerosols into the atmosphere through seawater bubble bursting (Monahan et al., 1986). In the high Arctic, where the ocean can be partially or fully covered by sea ice, additional polar-specific sources of sea salt emissions include blowing snow over sea ice (Yang et al., 2008; Huang and Jaeglé, 2017; Yang et al., 2019; Marelle et al., 2021; Frey et al., 2020) and sea spray from leads (Nilsson et al., 2001; Leck et al., 2002; Held et al., 2011; Kirpes et al., 2019). Studies have suggested frost flowers as another potential sea ice source of sea salt aerosol in Arctic coastal regions (Domine et al., 2004; Xu et al., 2016), although the extent to which they contribute is still uncertain (Huang et al., 2018; Kirpes et al., 2019). Sea salt aerosols generated through sublimation of saline blowing snow have been included in several atmospheric models (Yang et al., 2008, 2019; Huang and Jaeglé, 2017; Marelle et al., 2021; Gong et al., 2023). However, a dedicated parameterization for sea spray emissions from leads is not available. Leads are fractures in sea ice wide enough to be navigable by vessels (WMO, 2014). They typically have an elongated shape, with lengths from hundreds of meters to hundreds of kilometers, and widths from tens of meters to kilometers (Li et al., 2020). Sea spray in regions of leads are usually modeled identical to open ocean emissions, weighted by the open water fraction within the grid cell (e.g. Ioannidis et al., 2023), although it is known that leads have properties that make the emission process different from open ocean (e.g. Nilsson et al., 2001). The measured emissions over leads and open ocean show that there are (1) differences in the magnitude of sea spray particle emissions (number and mass), suggesting different driving mechanisms (Nilsson et al., 2001; May et al., 2016), and (2) differences in the composition of sea spray, with enhanced organic fraction when coming from leads (Leck et al., 2002; Kirpes et al., 2019). Therefore, models with a detailed representation of sea spray emissions from leads are needed to accurately represent their contribution to aerosol-radiation and aerosol-cloud interactions (Thornhill et al., 2021). In particular, in the warming Arctic as sea ice extent/thickness changes and the marginal ice zone (MIZ) becomes wider at certain times of year (Strong and Rigor, 2013), emissions from leads may be relatively more or less important than emissions from the open ocean or pack ice. Not including this process in models may

lead to an incomplete knowledge of the changing natural aerosol baseline in the Arctic, which may affect our representation of the Arctic climate.

There have been a limited number of dedicated measurements to quantify sea spray emissions from leads. Radke et al. (1976) measured sodium ($Na^+$) aerosol and CCN atmospheric concentrations at Utqiaġvik, Alaska, in March 1970, and found a correlation between lead openings near the station and increased $Na^+$ concentrations, but that lead occurrence had limited impact on CCN concentrations. Nilsson et al. (2001) measured aerosol turbulent fluxes over leads in the high Arctic in July–August 1996. They found (i) that emissions are larger over the open ocean than over leads by about one order of magnitude, (ii) that the size distribution of aerosols is dominated by a 2 µm diameter mode over leads, while there are two modes (100 nm and 1 µm) for open ocean emissions. Processes related to biological activity, the release of supersaturated gases, or sea ice melt were suggested to explain the formation and bursting of bubbles releasing sea spray in leads rather than whitecaps like over the open ocean. They also concluded that sea spray emissions from leads may be important for CCN populations in the Arctic. Leck et al. (2002) also suggested a non-wind-driven bubble bursting process, mostly active during sea ice melting days, consisting mainly of small-bubble-sourced jet drop mode emissions. This study also showed fractions of organics in sea spray from leads of up to 20% (in volume) in the high Arctic in July–August 1996. They also suggested that the increased organic fraction of sea spray reduced their CCN activity. Held et al. (2011) suggested that emissions from leads could explain 5–10% of the variations in measured particle number concentrations in August 2008 in the high Arctic. They also showed that elevated aerosol concentrations were more frequently observed over leads than over the continuous pack ice. May et al. (2016) showed that, over the period 2006–2009, the super-micron $Na^+$ aerosol mass can be multiplied by 4 when leads open near Utqiaġvik, Alaska, while sub-micron aerosols are more affected by long-range transport. This conclusion is based on a measured shift in mass distribution towards larger aerosol sizes when leads are present, and a greater influence of transported sea salt aerosol in a more continuous sea ice cover condition. Kirpes et al. (2019) demonstrated the need for inclusion of lead-sourced sea spray production in the modeling of Arctic atmospheric composition, and quantified the median organic fraction in emissions from leads between 30% to 50% at Utqiaġvik, Alaska, for 6 winter days in 2014. They further indicate that the observed organic coating could come from cryo-protectant gels produced by micro-organisms. At the same site and in spring 2016, Chen et al. (2022) also showed the important role played by leads both in sea salt aerosol direct emissions, and through deposition of sea spray onto snow which has the potential to be later re-suspended through blowing snow, although their measurements could not conclude on this last part.

In summary, sea spray emissions (mass and number) from leads (i) are probably comparable in magnitude but less than emissions from open ocean, (ii) might have higher organic contents than over open oceans, (iii) have a different size distribution than open ocean sea spray, with a dominant coarse mode. However, not enough information exists for both points (ii) and (iii) in order to be transferred directly for use in atmospheric emissions within models. Therefore, the parameterization proposed within this paper focuses on point (i) with the aim of exploring the possible range of sea spray emissions from leads at the scale of the Arctic. We also evaluate how this compares to the blowing snow source of sea salt aerosols and long-range transport of sea salt from lower latitudes to the high Arctic, in order to motivate future work refining the approach to account for points (ii)

and (iii). In order to better understand the role of leads in the Arctic aerosol budget, this paper addresses the following scientific questions:

- How can atmospheric models estimate sea spray emissions from leads from the current state of knowledge?
- What are the likely upper and lower bounds of sea spray emissions from leads in the Arctic?
- What is the modeled annual cycle of sea spray emissions from leads and how does it compare with blowing snow?
- What is the sensitivity of Arctic sea salt aerosol to sea spray emissions from leads?

To address these questions we first present a conceptual model of sea spray emissions from leads and the main methodological assumptions to formulate them for the Arctic Ocean in Section 2. Section 3 presents the sea spray emissions obtained from the conceptual model for the year 2018 and the sensitivity of these emissions to input parameters and compares the modeled sea spray emissions from leads with other sources of sea salt aerosol over the high Arctic Ocean (blowing snow and transport). The impact of this parameterization on atmospheric concentrations of sea salt is further evaluated using the Weather Research and Forecasting model coupled with Chemistry (WRF-Chem). The implications of this work are finally discussed in Section 4.

## 2 Materials and Methods

### 2.1 Methods

In this section we present the approach used for the parameterization of sea spray emissions from leads, including the measurements it is based on, the method to compute the inorganic and organic fractions of sea spray, and the parameterization of sea salt emissions from blowing snow for comparison with leads. The conceptual emission model built in this work is implemented in a series of Python3 Jupyter notebooks, available at Lapere (2024). In this work, the conceptual model is used to compute emissions for the year 2018.

#### 2.1.1 Lead flux parameterization

To our knowledge, the only detailed measurements of sea spray aerosol fluxes over leads that can readily be used for modeling purposes are found in Nilsson et al. (2001). They found that the total particle number aerosol flux over the open Norwegian Sea and Barents Sea was described by $F_{oo}$ (in $10^6\,\mathrm{m}^{-2}\,\mathrm{s}^{-1}$) in Equation 1, whereas over leads it corresponded to $F_{leads}$ (in $10^6\,\mathrm{m}^{-2}\,\mathrm{s}^{-1}$). In both cases, the 10 m wind speed (U in $\mathrm{m\,s}^{-1}$) appeared as the controlling factor. Unfortunately, the aerosol fluxes reported in Nilsson et al. (2001) are not size-resolved and cannot be used as-is in atmospheric aerosol models. Therefore, we propose to parameterize sea spray emissions from leads by applying a correction factor ($R_{Nilsson}$) to open ocean sea spray source functions, based on commonly used size-resolved sea spray source functions (described below). $R_{Nilsson}$ is calculated as the ratio between aerosol fluxes from open ocean and leads derived from Nilsson et al. (2001), which depends on 10 m wind speed (Equation 1 and Figure 1a).

$$R_{Nilsson} = \frac{F_{leads}}{F_{oo}} = \frac{e^{0.11[\pm 0.05]U - 1.93}}{e^{0.20[\pm 0.06]U - 1.71}} \tag{1}$$

The three sea spray source functions used in this work are Gong (2003) - GO03, Salter et al. (2015) - SA15, and Ioannidis et al. (2023) - IO23. GO03, together with Monahan et al. (1986), from which it is an adaptation, is the most commonly used open ocean sea spray source function in global climate models, such as the ones involved in CMIP6 (Lapere et al., 2023). On the other hand, SA15 departs from the usual whitecap approach, and includes a dependency on SST, which can be an important factor for polar oceans and for leads. Furthermore, the SA15 source function has been tested and validated against measurements at high latitude stations. Finally, the IO23 source function, implemented in the WRF-Chem model for Arctic studies also includes a dependency on SST but is based on a whitecap fraction approach, with a correction for smaller aerosols (see Section 3.7.1). By using these three functions (which formulations are described in Section A1), we can compare an approach commonly used in global models (Lapere et al., 2023) with more Arctic-adapted ones, including with a non-whitecap-based approach. The interested reader is invited to use the code of the conceptual model provided with this study to test any other source function. We choose aerosol diameters cutoff between 10 nm to 10 μm, typical of what most climate and atmospheric chemistry models use for the representation of sea salt aerosols. The size distribution of particles emitted from leads derived from applying $R_{\text{Nilsson}}$ to all three source functions is presented in Figure 1b. These source functions show different magnitudes for small-size aerosols, but similar ones for larger particles.

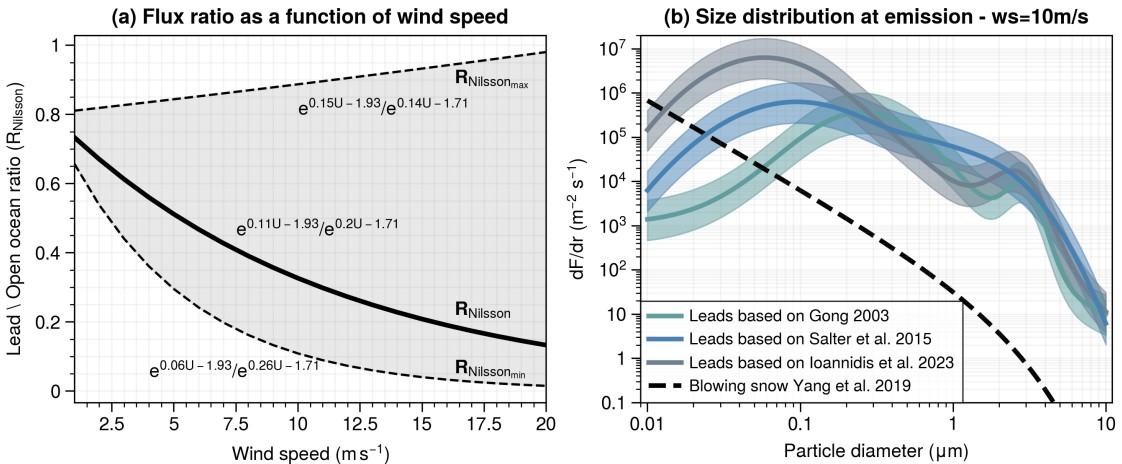

**Figure 1. Emission flux parameterization.** (a) Flux ratio derived from Nilsson et al. (2001) as a function of wind speed. Solid line is with the best fit coefficients from Nilsson et al. (2001), dashed lines are using minimum and maximum fit coefficients. (b) Size distribution of sea salt aerosol emissions from leads based on GO03 (green), SA15 for a 2°C SST (blue) and IO23 for a 2°C SST (grey), and blowing snow based on Yang et al. (2019) (dashed line), as implemented in this study. Shades are using the minimum and maximum $R_{\text{Nilsson}}$ coefficients.

Because emission fluxes from leads are uncertain, we also use the confidence intervals of the fitting parameters given by Nilsson et al. (2001), to derive the upper and lower possible boundaries for sea spray emissions from leads. The minimum and

maximum $R_{\text{Nilsson}}$ ratio based on these confidence intervals are described in Eq. 2,

$$R_{\text{Nilsson}_{\min}} = \frac{e^{0.06U-1.93}}{e^{0.26U-1.71}} \qquad \& \qquad R_{\text{Nilsson}_{\max}} = \frac{e^{0.15U-1.93}}{e^{0.14U-1.71}} \tag{2}$$

Figure 1a shows that $R_{\text{Nilsson}_{\max}}$ is greater than 0.8 for any wind speed, i.e. lead fluxes are similar to open ocean fluxes using
this ratio, and increases with wind speed. In contrast, $R_{\text{Nilsson}_{\min}}$ is less than 0.3 for wind speeds above $5\,\text{m}\,\text{s}^{-1}$ and declines
with increasing wind speed, similar to $R_{\text{Nilsson}}$.

### 2.1.2 Estimating the organic fraction of marine aerosols

Sea spray is generally composed of important fractions of organic material. Here, the organic fraction (OF) of sea spray
emissions is computed based on the work by Vignati et al. (2010), where the OF (in %) depends only on the chlorophyll-a
surface concentration in the ocean ($Chl$ in $\mu\text{g}\,\text{m}^{-3}$), as expressed in Eq. 3.

$$\text{OF} = 43.5\,Chl + 13.805\,, \text{ with limit values of 2\% and 76\%.} \tag{3}$$

For $Chl$, we use the daily average mass concentration of chlorophyll-a in sea water from the Global Ocean Biogeochemistry
Hindcast provided by the Copernicus Marine Service Information (Copernicus Marine Service Information, 2023a). Although
products such as Copernicus Marine Service Information (2023a) cannot be validated in leads due to the lack of sufficient num-
ber of in-situ data and model biases in under-ice phytoplankton phenology and sea ice coverage (Wakamatsu et al., 2022), this
is the best available option for our purpose. We tested several parameterizations for sea spray OF which connect chlorophyll-a
to OF with a simple relationship, including Vignati et al. (2010); Rinaldi et al. (2013); Gantt et al. (2015) (results not shown
here). We found that Vignati et al. (2010) yielded the most realistic values, which for the Arctic compared to measurements
from Leck et al. (2002) and Kirpes et al. (2019) (see Section 3.5). For consistency, the OF parameterization is used both in the
conceptual model (Sections 3.1 through 3.5) and in WRF-Chem (Section 3.7) to estimate the marine organic aerosol emissions.
As noted in the introduction, sea spray refers to the full emission flux (sea salt and OF). Sea salt is defined as the sea spray flux
minus its organic fraction (sea_spray*[1-OF]). The uncertainty on the Copernicus Marine Service Information (2023a) product
will therefore only affect the partitioning between inorganic sea salt and marine organics, but not the total sea spray emission
flux. In this work, the size distribution of organics is assumed to be the same as inorganic sea salt, which is a first estimate.
Future work that refines the size distribution of marine organic emissions from leads is needed, as measurements have shown
that inorganic sea salt and marine organics have different dominant modes (Prather et al., 2013; Wang et al., 2017), including
in the Arctic (Kirpes et al., 2019).

We acknowledge that there are limitations associated with parameterizing organic fraction using chlorophyll-a. However the
availability of chlorophyll-a products makes it more likely that these parameterizations can be adapted and used within climate
and Earth system models. While it is clear that other seawater characteristics than chlorophyll-a (such as organic carbon or
glucose concentration) can better account for the OF of sea spray (Fuentes et al., 2010; Quinn et al., 2014; Rocchi et al., 2024),
their lack of general availability in satellite or model-based products limits our ability to recommend these now for use by large
scale models.

### 2.1.3 Estimating sea salt aerosol from blowing snow

In addition to leads, blowing snow is an important source of sea salt aerosol in the Arctic, but the relative importance of blowing snow and leads for the Arctic sea salt aerosol budget is still an open question. In order to compare the emissions from leads obtained with our parameterization with sea salt emissions from blowing snow, we use the existing blowing snow emission parameterization described in Yang et al. (2008, 2019). The size-resolved fluxes from leads and blowing snow are compared in Figure 1b.

Although this blowing snow parameterization bears uncertainties, including snow salinity, number of sea salt aerosols released per snow grain, and fitting parameters (threshold wind speeds, surface temperatures, effect of snow age...), this work does not aim to investigate these uncertainties. The tuning parameters used here correspond to the ones used in Gong et al. (2023) and Confer et al. (2023) where more extensive validations against Arctic observations were conducted. In particular, Confer et al. (2023) showed that using variable or fixed snow salinity can change the total emission flux by 72%. As a result, the blowing snow fluxes are given as indicative for comparison with leads, but are not representative of the range of possible actual values, which are still uncertain and would require more observations to be estimated.

To evaluate where and when blowing snow events can occur, i.e. where there is enough fresh snow that can be mobilized, we consider the surface snow thickness from the daily Arctic Ocean Physics Reanalysis (Copernicus Marine Service Information, 2023b). If snow thickness on sea ice decreases from one day to the other, we consider that there has been melt or strong snow drift, and that the snow can no longer be mobilized. Blowing snow is considered to be able to occur otherwise. Although this assumption is a simplification, it is more realistic than the current state of modeling of blowing snow in atmospheric models, which usually assumes an infinite snow reservoir on sea ice. The difference between these two approaches (blowing snow source limited by decreased snow on sea ice cover vs snow on sea ice always available for lofting) is tested in Section 3.6.

## 2.2 Sea ice data, lead detection, meteorological data

### 2.2.1 Lead fraction derived from TOPAZ reanalysis

Sea ice concentration information in climate models and atmospheric chemistry models often relies on ocean reanalysis products such as the coupled ocean-sea ice Hybrid Coordinate Ocean Model-CICE-Tracers of Phytoplankton with Allometric Zooplankton (TOPAZ) (Sakov et al., 2012; Copernicus Marine Service Information, 2023b). This dataset provides daily Arctic sea ice area fraction, with a 12.5 km spatial resolution. Using this product, we assume that the lead fraction within a given grid cell is the fraction of open ocean, i.e. $(1 - \mathrm{sea\_ice\_concentration})$ whenever the sea ice concentration is above 80%. Below this threshold we consider that there are no leads. This value is chosen because it corresponds to the value that is sometimes used to define the transition between the MIZ and the pack ice (e.g. Vichi, 2022). The sensitivity to the selected threshold value is discussed in Appendix A. We note that this concept estimates the fraction of open water leads, which are the ones relevant for sea salt aerosol fluxes. In other contexts leads may be also covered by thin ice.

### 2.2.2 Lead fraction derived from NEMO-neXtSIM model

In addition to TOPAZ, we also use the sea ice concentration simulated by the coupled ocean-sea ice Nucleus for European Modelling of the Ocean-neXt generation Sea Ice Model (NEMO-neXtSIM) modeling system (Rampal et al., 2016; Boutin et al., 2023, 2024). NEMO-neXtSIM differs from TOPAZ in various aspects. In particular, this model includes a brittle rheology (the Bingham Maxwell rheology of Ólason et al. (2022)) to simulate the sea ice mechanical response to winds and ocean currents. It was shown capable to simulate very similar sea ice deformation statistics and scaling properties compared to what is found in satellite observations (Rampal et al., 2019; Ólason et al., 2022), in particular those driving the formation of leads (Ólason et al., 2021).

This sea ice concentration data set is referred to as neXtSIM hereafter (Boutin et al., 2024). The outputs originally consist of 6-hourly fields that are given on a regular grid of 12-km horizontal spatial resolution for the high Arctic region. However, we work with the daily-averaged fields computed from these original files. We define leads in this simulation the same way as in the TOPAZ model, i.e. as the fraction of open ocean where sea ice concentration is larger than 80%.

### 2.2.3 Lead fraction from satellite detection

Satellite-derived lead detection products are also available, such as Willmes et al. (2023a) which provide pan-Arctic daily maps of lead detection, based on thermal contrast from the Moderate Resolution Imaging Spectroradiometer (MODIS) data. This product is hereafter referred to as ArcLeads. The spatial resolution of ArcLeads is 1 km, and the time period covered is November to April for the years 2002 to 2021. In their product, Willmes et al. (2023a) classify data in grid cells according to 6 categories (cloud, land, sea ice, artefact, lead, open water). This data is only available for cloud free cases thus there can be gaps in the daily dataset. Leads in this dataset are also not necessarily ice free. According to the WMO sea ice nomenclature definition leads can have ice up to 30 cm thickness (WMO, 2014). The exact upper ice thickness limit for the ArcLeads product is not known. As ice covered leads will not allow sea spray aerosol fluxes these aspects need to be considered in the interpretation.

### 2.2.4 Spatial processing of sea ice data

The lead fraction we consider from models is not strictly a lead fraction as it includes all areas of open water within the pack ice where the sea ice concentration is greater than 80%. In coastal regions and MIZ, it is challenging to differentiate leads from other open water areas such as e.g. polynyas (Ólason et al., 2021). In order to avoid falsely computing emissions from leads in these regions, a mask excluding coastal regions and MIZ is created and applied on all surface products to ensure we only consider regions where the confidence in lead modeling/detection is higher (black contour in Figure 2). The area inside this contour is referred to as the high Arctic. We acknowledge that there are also actual leads outside the contour depending on season, especially in the marginal seas. For a conceptual understanding of sea spray aerosol emissions from leads, however, we chose to focus only on the high-confidence area for lead detection/modeling.

Furthermore, all data sets are regridded onto the TOPAZ grid which has the lowest resolution. For the ArcLeads satellite product, the number of 1 km grid cells which are flagged as lead within each 12.5 km TOPAZ grid cell, are divided by the total number of ArcLeads cells within the cell. This computation yields the ArcLeads lead fraction inside each TOPAZ grid cell. Leads are relatively small scale structures, with typical width between 50 m up to kilometers (Li et al., 2020), so the resolution of 12.5 km used here is not ideal for reproducing individual leads. However, the objective of this work is not to investigate the behavior of individual leads, but to estimate emissions from leads at the Arctic scale. In addition, the target application of the emission model provided in this work is regional- to large-scale models, which are seldom run below resolutions of 10 km for polar studies. Therefore, the 12.5 km TOPAZ resolution is appropriate for both the purpose of this work and its foreseen applications.

### 2.2.5 Meteorological data

In order to calculate the sea spray and blowing snow emissions, meteorological data is needed. In the conceptual model, we use the 10 m wind speed and 2 m air temperature from the ECMWF Reanalysis v5 (ERA5) hourly reanalysis (Copernicus Climate Change Service, 2017), which we re-sample to a daily frequency. Additional data sets used in this work also include the Modern-Era Retrospective analysis for Research and Applications Version 2 (MERRA-2) aerosol reanalysis (Global Modeling and Assimilation Office (GMAO), 2015) to estimate the relative importance of lead and blowing snow sea salt emissions compared to transport from the lower latitudes (i.e. northward transport through the high Arctic contour in Figure 2). Emissions of sea salt in MERRA-2 are computed online at hourly resolution (Randles et al., 2017). Although no validation of MERRA-2 sea salt aerosol is carried out in this work, previous studies on polar aerosols have relied on MERRA-2 data and showed reasonable performance of this product at high latitudes (Xian et al., 2022; Zamora et al., 2022; Böö et al., 2023). However, Lapere et al. (2023) found a large positive bias in sea salt aerosol surface concentrations in MERRA-2 compared to Arctic stations. Therefore, the sea salt aerosol transport computed here is probably an overestimation. Observations of $Na^+$ atmospheric concentrations at Arctic stations are taken from Norwegian Institute for Air Research (2024) and Yang et al. (2019), and planetary boundary layer heights from Esau and Sorokina (2011). The results presented in this work are all for the year 2018.

## 3 Results and discussion

### 3.1 Sensitivity to the choice of sea ice information

In order to evaluate how critical the surface information is for estimating sea spray from leads, we compute emissions in the conceptual model using the three sea ice data sets described in Section 2.2, and present them in Figures 2 and 3. Throughout this section, the fluxes presented are the average of the three source functions (GO03, SA15, IO23), in order to isolate the impact of the choice of the sea ice data set, all other things remaining equal. Comparisons between the source functions are presented from Section 3.2 onwards.

Using ArcLeads results in lead fractions and emission fluxes more than one order of magnitude larger than with TOPAZ and neXtSIM. This discrepancy can be explained by the detection method in ArcLeads, which is based on thermal contrast from MODIS ice surface temperature data (Reiser et al., 2020). First, the contrast is not strong enough in summer months, hence the absence of detection data between May and October. Second, leads covered with a thin layer of ice still have a contrast strong enough so that they are detected, whereas we only want to consider the open water leads for sea spray emissions in the model. In this regard, the ArcLeads product does not provide the fraction of open water leads at a given moment, but rather the fraction of both currently open leads and leads that opened in the previous days that are now covered by thin ice, as the freezing and opening of leads are processes that are faster than the timescale of the satellite observations. This is different from what we extrapolate using the sea ice fraction from TOPAZ/neXtSIM, which is the daily average fraction of open water in the pack ice. This explains why the lead fraction from ArcLeads is so much larger than the lead fractions we derive from TOPAZ and neXtSIM. Leads by definition can be covered by thin ice (WMO, 2014) and that is what ArcLeads provides. But for our study we only need open water leads. Thus, this means that using ArcLeads provides an overestimation of the upper bound of sea spray fluxes from open leads, probably by one order of magnitude, and so ArcLeads is not usable as is for our sea spray aerosol parameterization. Lead fraction can vary by one order of magnitude in observational lead products depending on the considered physical lead properties, as reported by von Albedyll et al. (2024). There are satellite products that can provide open water fraction based on ice divergence (e.g. von Albedyll et al., 2024), but they are currently not available on the spatio-temporal scales needed for our study.

Generally, the same spatial distribution of areas of higher sea spray emissions from leads (calculated using the average of the three source functions) appear with the three products for the winter and spring months as shown in Figure 2, except in the Beaufort Sea where ArcLeads yields a maximum of lead fraction (around 20% on average - Figure 2c) and therefore an emission hot spot that is not present with the other two products (Figure 2d,e,f). The greater lead fractions detected in the Beaufort Sea are discussed in Willmes et al. (2023b) and attributed to stronger wind divergence in this area, along with thinner sea ice which leads to more frequent breakup events in winter (Rheinlænder et al., 2024). Both ArcLeads and neXtSIM display patterns typical of leads, showing narrow elongated shapes of increased fluxes, as opposed to the TOPAZ reanalysis which displays a more homogeneous field without discernible lead shapes. In this regard, neXtSIM is more appropriate than TOPAZ for detecting leads based on sea ice concentration. Therefore, neXtSIM is the sea ice data set that will be used in the next sections to compute sea spray fluxes from leads.

Major differences in the magnitude of emissions between the three sea ice products are also obtained each month when aggregated at the regional level, as shown in Figure 3. A similar seasonal cycle and magnitudes are obtained in TOPAZ and neXtSIM but larger magnitudes are given by ArcLeads. In winter, using ArcLeads generates an emission flux 10 to 15 times larger than with TOPAZ or neXtSIM, albeit with similar variations from month to month (Figure 3). For this period, TOPAZ and neXtSIM produce similar fluxes, although TOPAZ produces on average more sea salt aerosol. Using both of these model products gives the same seasonality, with maximum emission fluxes between July and September, and minimum emissions in March and April, close to two orders of magnitude smaller than in summer. In addition, using $R_{\text{Nilsson}_{\min}}$ or $R_{\text{Nilsson}_{\max}}$ for

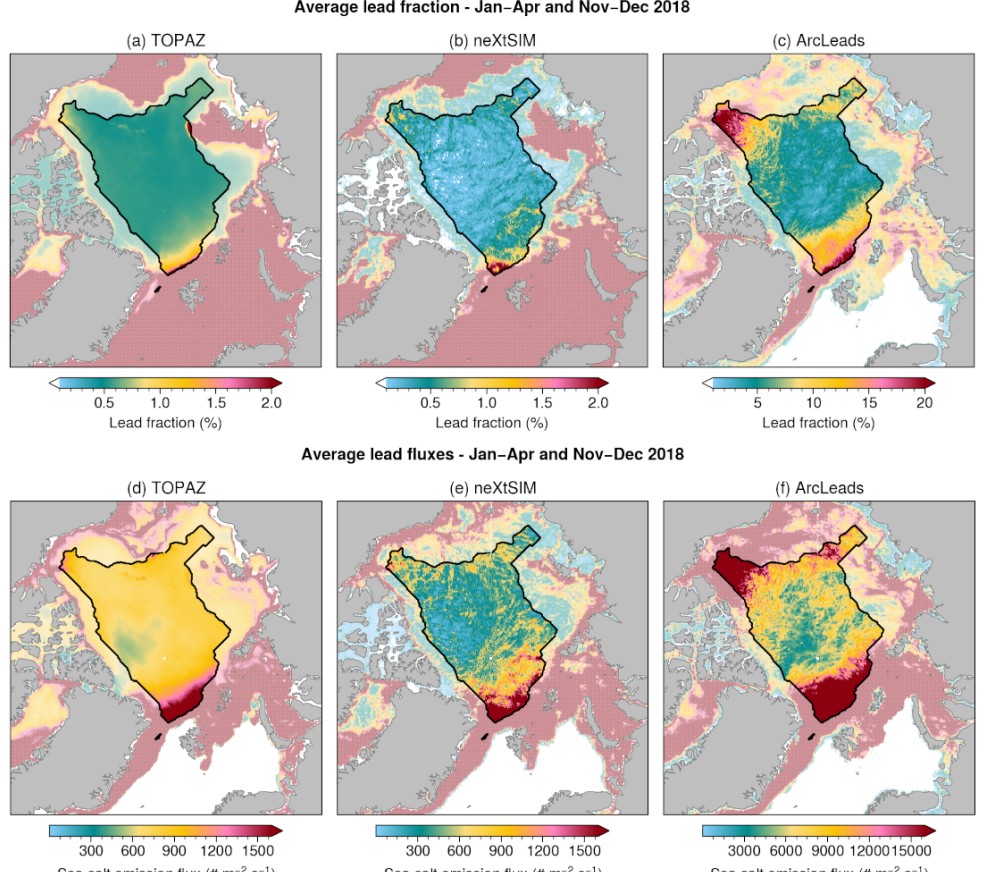

**Figure 2. Winter-Spring lead fractions and sea salt emissions.** Average lead fraction in (a) TOPAZ, (b) neXtSIM and (c) ArcLeads, for Jan–Apr and Nov–Dec 2018. The black contour indicates the area referred to as the high Arctic in this work. Values outside the high Arctic contour are provided for information but are not considered as fluxes from leads. (d) Average sea salt particle number emission flux from leads for Jan–Apr and Nov–Dec 2018, using the average of the three source functions applied to TOPAZ, neXtSIM (e) and ArcLeads (f). Note: the color scale is different in panel (c) and panel (f) compared to (a,b) and (d,e), respectively.

the flux computation (error bars in Figure 3) can result in a difference of up to one order of magnitude in the total particle number flux in the high Arctic.

## 3.2  Emissions from leads versus blowing snow

Emission of sea salt through blowing snow is known to be an important source of aerosols in sea ice regions (Yang et al., 2008, 2019; Frey et al., 2020; Gong et al., 2023). However, no regional comparison of the relative magnitude of sea salt aerosol
fluxes from blowing snow and leads exist, to date. In Figure 4 we compare, for the first time, the seasonal spatial distribution

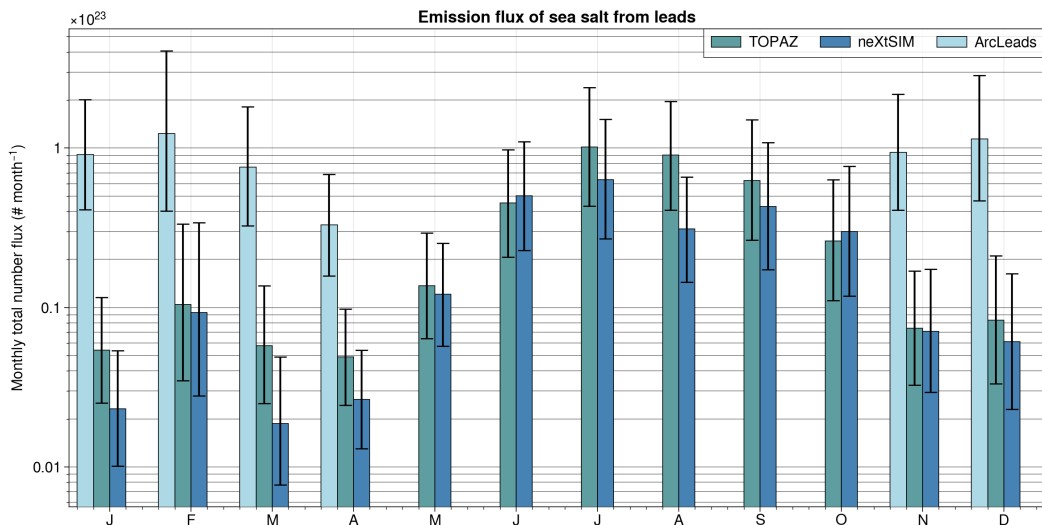

**Figure 3. Annual cycle of sea spray emissions from leads.** Monthly sum of sea salt aerosol number emissions from leads over the high Arctic using sea ice information from the satellite detection from ArcLeads (light blue), TOPAZ sea ice reanalysis (cadet blue) and sea ice modeling from neXtSIM (steel blue). Error bars indicate the particle number flux obtained using $R_{\mathrm{Nilsson_{min}}}$ and $R_{\mathrm{Nilsson_{max}}}$ instead of $R_{\mathrm{Nilsson}}$. Data for the year 2018 averaged for the three source functions. Note: the y axis uses a logarithmic scale.

of the sea salt particle number emission flux from blowing snow and leads. In winter, spring and fall, blowing snow in the high Arctic produces sea salt particle number emissions on average 1 to 2 orders of magnitude larger than emissions from leads.

The latter are more heterogeneously distributed, and are larger in the area between Greenland, Svalbard and the North Pole. In summer, sea salt number emissions from leads increase and become equivalent to blowing snow emissions, which are at their annual low, one order of magnitude lower than the other seasons. For that season the spatial distribution of sea spray emissions from leads also differs from the other seasons. Figure 4 also shows that the areas of maximum emission fluxes for blowing snow and leads are not found in the same regions, which is an important characteristic for the pan-Arctic aerosol budget.

The comparison is further studied by aggregating the emissions over the area defined as the high Arctic (see the black contour in Figure 4). Despite blowing snow dominating the sea salt particle number emissions throughout most of the year (except summer - Figure 4 and 5a), fluxes from leads in terms of mass emission of sea salt aerosol can be up to 2 orders of magnitude larger than blowing snow, especially in the summer months (Figure 5b). The difference between number and mass obtained here is connected to the blowing snow parameterization releasing mostly fine mode $Na^+$ aerosols in large quantities but significantly less coarse mode aerosol, and emissions from leads, in our parameterization, being much higher for particles bigger than 100 nm (Figure 1b). Using the IO23 source function results in particle number and mass fluxes larger than with the GO03 and SA15 source functions, consistent with the size distributions presented in Figure 1b.

Summed over the whole year, sea salt emissions from leads represent between 0.3% (with $R_{\mathrm{Nilsson_{min}}}$ and GO03 source function) up to 9.8% (with $R_{\mathrm{Nilsson_{max}}}$ and IO23 source function) of the sea salt number emissions in sea ice regions (i.e.

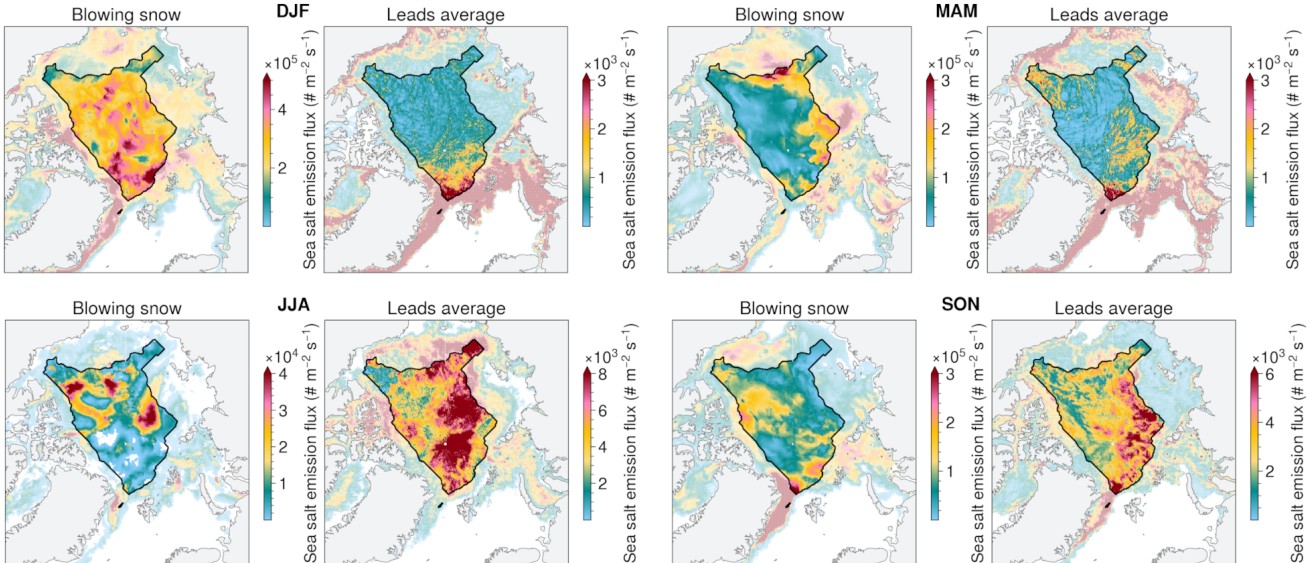

**Figure 4. Seasonal emissions from leads and blowing snow.** Average particle number emission flux of sea salt aerosol from blowing snow and leads for each season (DJF: December–February, MAM: March–May, JJA: June–August, SON: September–November). The black contour indicates the area defined here as the high Arctic. The average of the three source functions is presented, using $R_{\mathrm{Nilsson}}$ and neXtSIM sea ice concentration for lead definition. Note: color scales are different from one panel to the other to better show the spatial patterns. Color scales for blowing snow go 1 to 2 orders of magnitude higher than for leads.

blowing snow+leads), with mean values (i.e. with $R_{\mathrm{Nilsson}}$) of 0.7% to 4.5% depending on the source function. For the sea salt aerosol mass emissions, leads account for a greater fraction, between 30% and 85% depending on the parameterization, with mean values of 51% to 72%. The greater importance of blowing snow for number emissions and leads for mass emissions is consistent with previous observations, which revealed mostly coarse size sea salt aerosols from leads (Nilsson et al., 2001; May et al., 2016) and mostly fine mode sea salt aerosols from blowing snow (Frey et al., 2020; Gong et al., 2023). For cloud-related studies, models using a one-moment cloud microphysics scheme (based on aerosol mass only - e.g. Kessler (1969)) will therefore be more sensitive to emissions from leads, but two-moment schemes (aerosol mass and number - e.g. Morrison et al. (2005)) will likely find that sea salt from blowing snow plays a major role.

### 3.3 Lower and upper bounds of emissions from leads

With the approach and numbers presented above, we can bound the annual mass of sea salt aerosol emitted from leads into the high Arctic atmosphere for the year 2018 between 0.1 and 2.1 Tg yr$^{-1}$ across the three source functions and two sea ice data sets (Figure 6). For blowing snow, we find a total sea salt aerosol emission flux of 0.2–0.3 Tg yr$^{-1}$ for this region, which is consistent with the results of Confer et al. (2023) who found 0.28–2.24 Tg yr$^{-1}$ for a larger region of the Arctic. In comparison, MERRA-2 gives, for 2018, a mass of sea salt aerosol transported into the high Arctic of 1.8 Tg yr$^{-1}$. This last

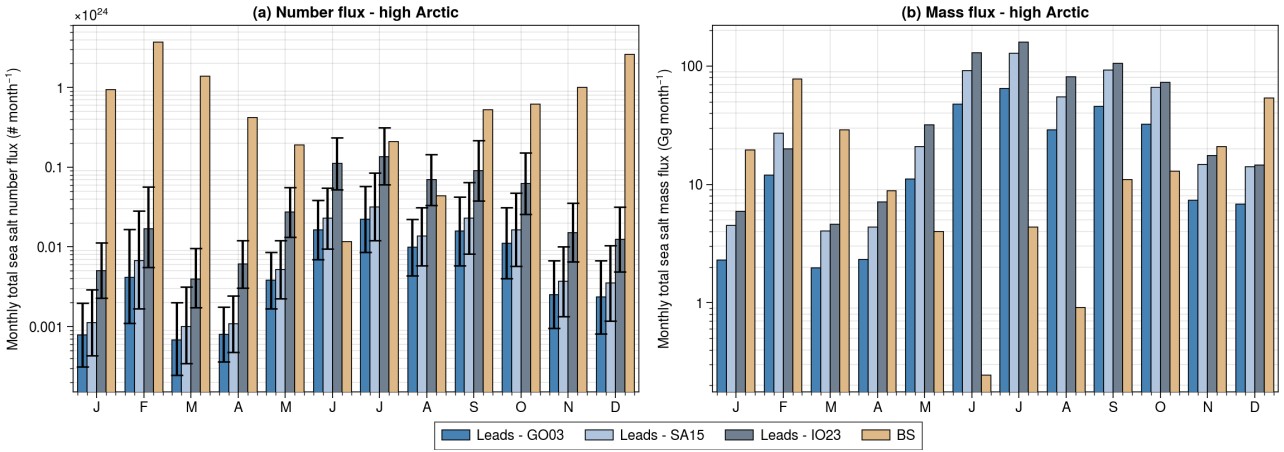

**Figure 5. Seasonality of emissions from leads and blowing snow.** (a) Monthly sum of sea salt particle number emission fluxes integrated over the high Arctic as defined by the black contour in Figure 4. Steel blue is leads using the GO03 source function, light blue is with the SA15 source function, gray is with the IO23 source function, and brown is blowing snow. Error bars indicate the particle number flux obtained using $R_{\mathrm{Nilsson_{min}}}$ and $R_{\mathrm{Nilsson_{max}}}$ instead of $R_{\mathrm{Nilsson}}$. All values are using the neXtSIM sea ice concentration for lead definition. Data for the year 2018. (b) same as (a) but for mass fluxes.

number is obtained by integrating the positive component of the meridional total column sea salt aerosol mass flux (*SSFLUXV*) from MERRA-2 hourly data in 2018 (Global Modeling and Assimilation Office (GMAO), 2015), along the boundaries of the high Arctic mask shown in Figure 4. Within these annual totals, the northward transport into the high Arctic is more important

325 in winter months, whereas leads gain importance in the summer months (Figure 6). Blowing snow shows a similar seasonality as transport overall, although with smaller sea salt mass fluxes, around one order of magnitude less (Figure 6). Therefore, sea salt emissions from leads - and blowing snow - may be almost as important as sea salt transported from the open ocean for the aerosol mass budget in the high Arctic.

 The uncertainty in sea salt emissions from leads is large, as illustrated by the blue shade in Figure 6. We evaluate the

330 sensitivity of the sea salt emissions from leads in the high Arctic (both mass and number) to each assumption made in the conceptual model (sea ice data - TOPAZ or neXtSIM, $R_{\mathrm{Nilsson}}$ ratio - average, min or max, source function - GO03, SA15 or IO23). For a given source function and sea ice product, using $R_{\mathrm{Nilsson_{max}}}$ versus $R_{\mathrm{Nilsson_{min}}}$ results in a change of the total flux by around 132–151%. In comparison, changing the source function, for a given $R_{\mathrm{Nilsson}}$ and sea ice product, results in a 22–26% difference in the total flux between IO23 and SA15 and 85–87% between IO23 and G03. Finally, for a given source

335 function and $R_{\mathrm{Nilsson}}$, using TOPAZ versus neXtSIM induces a 35–39% change in the total sea salt aerosol flux.

 As explained in Section 3.1, emissions from leads computed using ArcLeads are very different from the other two products because ArcLeads includes leads covered by thin ice. The differences between the fluxes obtained with ArcLeads and with the model-based products therefore originate from a different definition of what a lead is, rather than uncertainty in the models or the satellite product, and ArcLeads most likely provides an unrealistic upper bound of sea spray fluxes from leads. Therefore,

within our proposed parameterization the largest uncertainty for bounding the sea spray emissions from leads in the Arctic is
currently due to the uncertainty/scarcity of emission flux measurements.

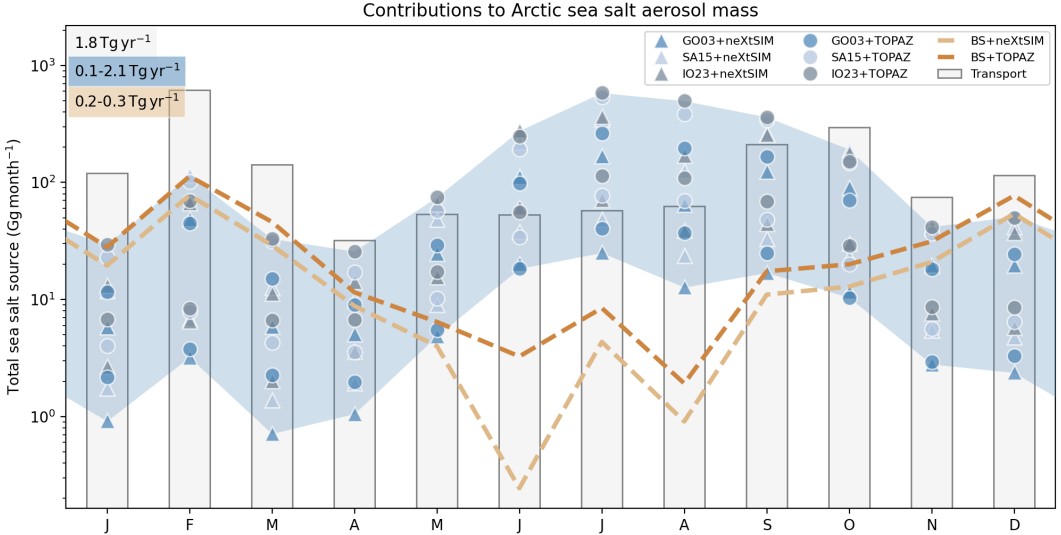

**Figure 6. Transport and local production of sea salt aerosol.** Monthly total emission and transported sea salt aerosol mass in the high
Arctic. Light gray bars show transported sea salt aerosol derived from MERRA-2, blue shades indicate the emission flux from leads using
$R_{\text{Nilsson}_{min}}$ (lower bound) and $R_{\text{Nilsson}_{max}}$ (upper bound) for the three source functions and two sea ice data sets, brown dashed lines
indicate the emission flux from blowing snow using the two sea ice data sets. Annual totals are shown in text boxes. Data for the year 2018.

## 3.4    Impact of emissions from leads on the annual cycle of sodium aerosol concentration

Climate models generally do not explicitly consider sea spray emissions from leads or sea salt from blowing snow. Instead,
they either ignore sea spray emissions from sea ice regions, or apply the open ocean source function in every grid cell, weighted

by the open water fraction. In what follows, this approach is referred to as the *usual approach* and is compared to the approach
including sources from leads and blowing snow, with a view to compare the annual cycle of sea salt aerosol obtained with both
approaches. In Figure 7, we show that by replacing the *usual approach* (dashed black line), with dedicated lead and blowing
snow parameterizations (filled curves), a better representation of the seasonal cycle of atmospheric concentrations of Na$^+$
aerosol can be obtained compared to observations (gray bars) at two high latitude stations affected by sea ice (Alert station,

Canada (82.49°N, 62.34°W) and Utqiaġvik station, Alaska (71.2°N, 156.0°W)). Observations at Alert include aerosol sizes up
to 1 μm, while Utqiaġvik is total suspended particles. In this Section, we remove the condition of being inside the high Arctic
mask of Figure 2 and consider that leads can be found wherever sea ice concentration is greater than 80%, including in coastal
areas near the two stations considered. Although the validity of such an approach needs further investigation, it is necessary
here as only coastal stations provide year-long observations with a robust annual cycle. Furthermore, grid points with sea ice

concentration below 80% are ignored, since the objective of this part is to assess whether sea ice sources only can explain the seasonal variations of sea salt aerosols for these locations.

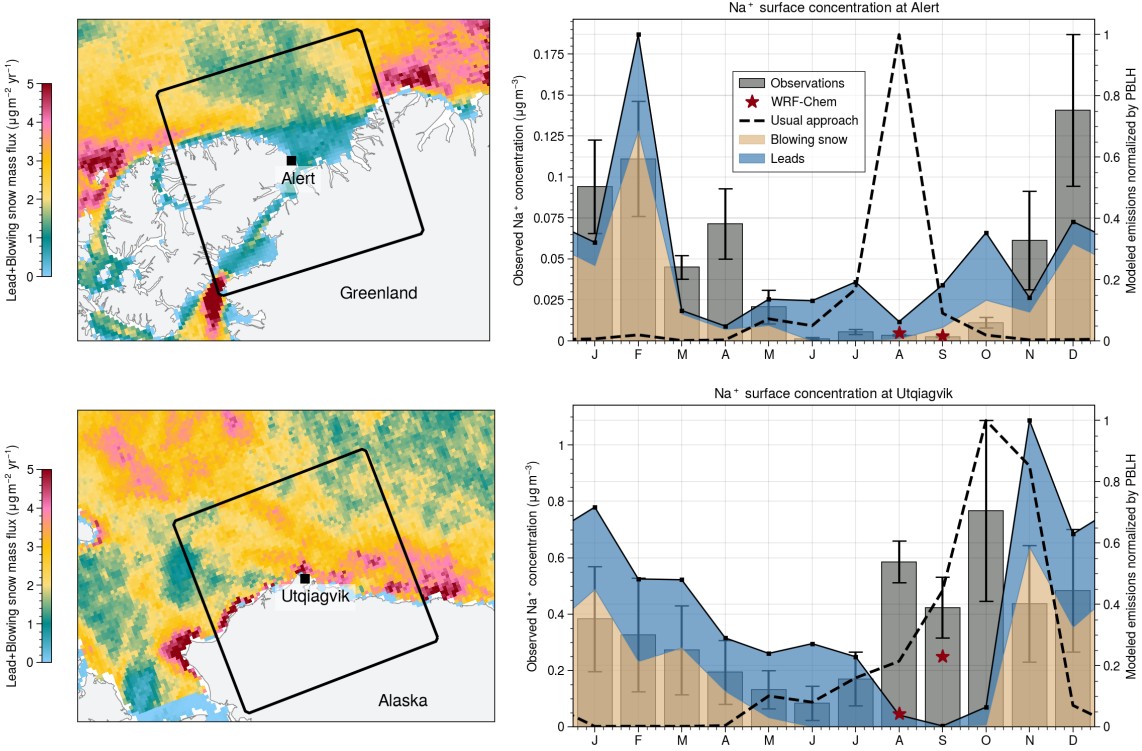

**Figure 7. Annual cycle of concentrations at Alert and Utqiaġvik.** Left: annual sea salt aerosol mass flux from blowing snow and leads near Alert (top) and Utqiaġvik (bottom) station (black square). The black contour indicates the area used for averaging fluxes in the right panel. Right: Monthly mean concentration of $Na^+$ aerosol at Alert (top) and Utqiaġvik (bottom) station from observations (Norwegian Institute for Air Research (2024), Yang et al. (2019) - gray bars - error bars show one standard deviation) and simulated in WRF-Chem (red stars - see Section 3.7), and proxy for concentrations using the open ocean flux applied to the open water fraction only (dashed black line) or blowing snow+leads (filled curves). All time series are normalized by their maximum for readability. For the model, emissions are divided by the boundary layer height values from Esau and Sorokina (2011) to mimic the concentration annual cycle. All fluxes are computed using the mean of the three source functions and average $R_{\mathrm{Nilsson}}$.

At Alert, the *usual approach* results in a reversed seasonality compared to observations, with $Na^+$ aerosol concentrations peaking in summertime, while observations show a wintertime maximum. However, when considering blowing snow emissions, a wintertime peak in $Na^+$ aerosol concentration is obtained, more consistent with observations. Even considering these
360 sea ice sources of sea salt, the seasonality still does not match the observations, which signals that our parameterization could be improved despite already showing better agreement than the *usual approach*. Also according to our model, leads at Alert

during winter seem to play a minor role on mass concentrations compared to blowing snow, while during the summer months leads dominate the mass concentration. At Utqiaġvik, observations show maximum $Na^+$ aerosol concentrations in August–October, consistent with open ocean sourced sea salt aerosol, but significant values are also observed in winter months, which the *usual approach* does not capture (dashed line is mostly zero outside of September–November). When considering leads and blowing snow, the seasonality in December–July matches very well the observations. For this location, we find that leads and blowing snow emit similar masses of $Na^+$ aerosol, although lead emissions are found during a longer period in the year.

Although the method used here to compute the proxy for concentrations has major limitations as it does not account for transport nor removal (wet and dry), the annual cycle obtained with the *usual approach* is similar to the annual cycle of concentrations yielded by climate models from CMIP6 (which use the *usual approach* for emissions) at these two locations, as shown in Lapere et al. (2023). Therefore, despite the important simplifications in Figure 7, it strongly suggests that local sea ice sources of sea salt aerosol need to be accounted for to obtain a decent annual cycle of $Na^+$ concentrations in the coastal Arctic. The absence of sea salt aerosol sources in sea ice regions in climate models was discussed in Lapere et al. (2023) as one reason why CMIP6 models do not compare well with the annual cycle of aerosol observations at high latitude. With the proxy calculations in Figure 7 we show that this is indeed likely to explain this current model shortcoming.

### 3.5 Marine organic fraction of sea spray from leads

Measurements in the high Arctic indicate an important fraction of organic material (OF) in sea spray from leads (Leck et al., 2002; Kirpes et al., 2019), which can affect the ability of sea spray to form liquid droplets (CCN) or ice crystals (INP), as more organics favors INP conversion over CCN. In what follows, we use the OF parameterization from Vignati et al. (2010) to estimate the OF in sea spray emitted from leads (see Section 2) and compare with observations. The seasonal variations of sea spray OF at the Arctic scale obtained with this parameterization are shown in Figure 8. Over the open ocean, the OF varies between 20% in wintertime to a maximum of 45% in springtime, consistent with usually observed phytoplankton blooms at that season in the North Atlantic (Cole et al., 2015). In contrast, in sea ice regions the OF is smaller, with a minimum in wintertime, around 17%, and peaks in the fall at 30% on average. In September, the OF in sea spray from leads is similar to the one in the open ocean. Annually, we estimate that organic material emissions from leads are comprised between 0.02 to $0.35\,\mathrm{Tg\,yr^{-1}}$ depending on the choice of source function and $R_{\mathrm{Nilsson}}$ ratio. The similar magnitude and complementary seasonality of primary marine organics from the open ocean and from leads is an important characteristic that should be further investigated as it could be critical for cloud nucleation and cloud phase. However, this mass estimate is uncertain because, contrary to the findings from measurement campaigns, we assume here the same size distribution at emission for leads and open ocean, and for inorganic sea salt and organic material.

The OF values obtained for 2018 in our model are comparable with measurements from past Arctic expeditions. Leck et al. (2002) conducted measurements in the high Arctic during the 1996 Arctic Ocean expedition (AOE-96), including of concentrations of sea spray found in the atmosphere near leads. They estimated the OF of the measured sea spray to 20% in July–August. Although our data is for the year 2018, we find a consistent OF in sea spray emissions from leads, between 16%–18% in July–August (Figure 9 top row). Similarly, Kirpes et al. (2019) conducted measurements near Utqiaġvik, Alaska,

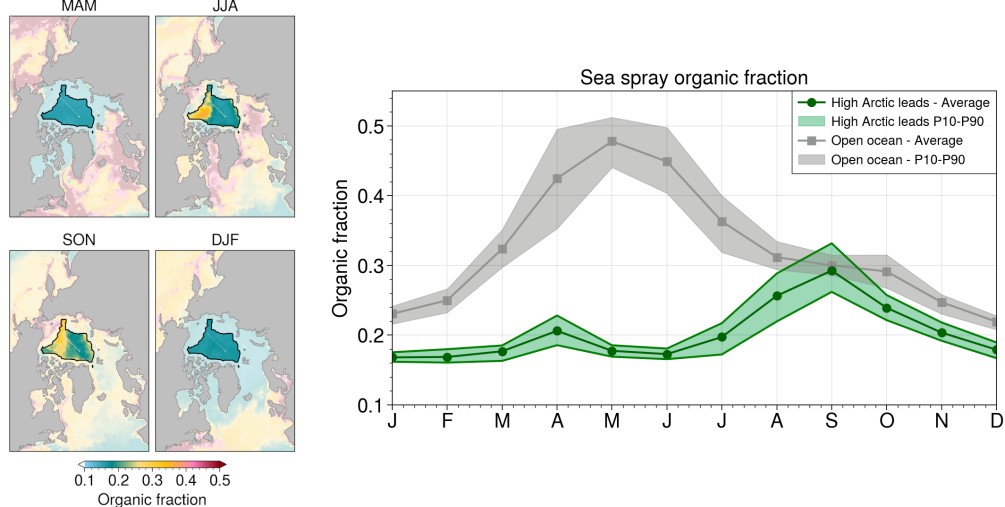

**Figure 8. Sea spray organic fraction in the Arctic.** Organic fraction in sea spray emissions based on the Vignati et al. (2010) parameterization and Copernicus Marine Service Information (2023a) chlorophyll-a concentration. The gray line is averaged over the open ocean areas (i.e. all areas where sea ice concentration is zero within the domain represented in the maps), weighted by the emission flux. The green line is averaged over lead emission areas in the high Arctic, weighted by the emission flux. The associated shades show the 10[th] and 90[th] percentiles over the corresponding area.

in February 2014, and estimated OF of 30% to 50% in sea spray near leads. In our case, for February 2018, we find an OF of 16% at this location (Figure 9 bottom row).

One possible explanation for the discrepancy between the OF we model and the measurements by Kirpes et al. (2019) is that their observations may include some transported / aged sea spray, and not only freshly emitted aerosols. Figure 8 shows that we predict a higher OF for open ocean sea spray than for leads during the season of the measurements. Therefore, if the conceptual model considered not only leads but also transported open ocean sea spray, we would obtain a higher OF, closer to the measurements. Furthermore, uncertainties also arise from the chlorophyll-a product as there is not enough observational data and there are model biases in both under-ice phytoplankton phenology and sea ice coverage (Wakamatsu et al., 2022). Additionally, massive phytoplankton blooms have been shown to occur under sea ice (Arrigo et al., 2012), which are likely missed by the model since it does not account for light availability under sea ice. Therefore, the use of modeled chlorophyll-a data has significant uncertainties, even more so for its use in leads.

## 3.6 Sensitivity of the parameterization

In the process of deriving the parameterization for sea spray emissions from leads proposed in this work, several assumptions are made. Hereafter, we test how sensitive the resulting emissions are to these assumptions.

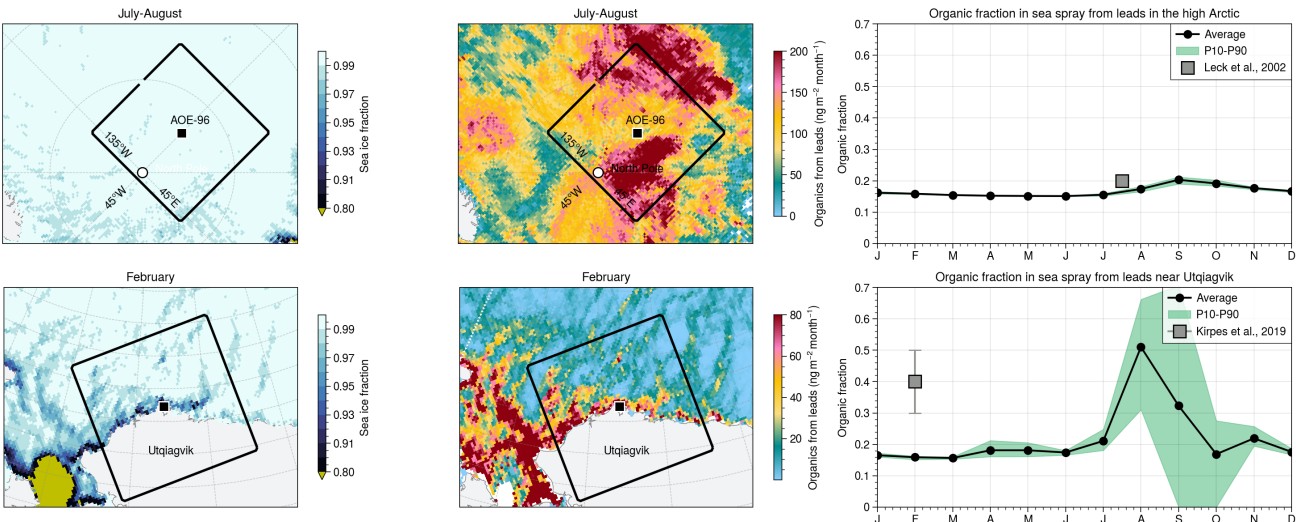

**Figure 9. Organic fraction compared to measurements.** Top left: Sea ice fraction in the central Arctic in July–August 2018. Top middle: Mass flux of organic aerosols from leads in July–August 2018. The black square marks the location where the measurements were made. Top right: Seasonal cycle of organic mass fraction in sea spray from leads averaged over the black contour in the middle panel (black line) and associated 10th and 90th percentiles over the area (green shade). Grey square is the OF measured in July–August 1996 by Leck et al. (2002). Bottom row: same as top row but for Utqiaġvik station and data from February 2014 by Kirpes et al. (2019).

Following the definition of the marginal ice zone given in Vichi (2022), we define leads as open water areas within the pack ice when sea ice concentration is above 80%. Figure A1 shows the monthly sea salt emissions from leads if, instead of 80%, the threshold on sea ice concentration is taken at 70% or 90%. In the period from November to April, the aggregated emissions in the high Arctic are not sensitive to the threshold choice, since sea ice concentration is higher than 90% almost everywhere in the domain. The biggest change is observed in the summer months (June–September), when choosing the more restrictive threshold (90%) yields an important decrease of emissions, by a factor of 5 in July–August. For that same period, lowering the threshold at 70% increases the emissions compared to 80%, but less than the decrease obtained with the 90% threshold. Consistent with the previous results, the sea salt aerosol mass emissions are less sensitive to the threshold than the sea salt aerosol number emissions. Importantly, irrespective of the threshold for leads, blowing snow remains the dominant source in terms of sea salt number emissions throughout the year, and leads remain the dominant sea salt mass emission source in summer months.

For simplicity, only the year 2018 is considered throughout this work for the analysis and evaluation of the parameterization for sea spray emissions from leads. We test the year-to-year variability by computing emissions for the year 2017 as well. Figure A2 shows that the absolute values, seasonal variations, and comparison between leads and blowing snow are similar for both years. In particular, the total sea salt aerosol mass emission for the high Arctic, using the Gong (2003) source function with $R_{\mathrm{Nilsson}}$ ratio, is 0.26 Tg (0.24 Tg, respectively) for leads (blowing snow, respectively) in 2018, and 0.20 Tg (0.26 Tg,

respectively) in 2017. This corresponds to a 31% increase for leads, and 6% decrease for blowing snow, between 2017 and 2018. Although the totals are not very different, these variations are still substantial, which indicates that sea spray from leads can account for an important fraction of the inter-annual variability in sea salt mass concentration in the high Arctic.

In Section 2.1.3, an assumption is made on snow thickness variations to determine whether blowing snow events can occur. Removing this condition and assuming an infinite snow reservoir instead results in emissions around 45% larger, both for sea salt number and mass (Figure A3), except in June. For that month, the emissions increase by one order of magnitude. Although the approach for determining the blowing snow reservoir adopted in this work needs further refinement, we show that it is not critical for the sea salt aerosol emissions from blowing snow, except in summer.

Finally, our work relies on the total aerosol flux measurements by Nilsson et al. (2001). In order to compensate using only one set of measurements, we explore the whole range of uncertainties from the regressions given by Nilsson et al. (2001), including the confidence interval of the fit coefficients. Importantly, Nilsson et al. (2001) also report measurements of open ocean total aerosol fluxes (as described by the denominator in Equation 1). These are similar in magnitude and wind dependence as the total fluxes obtained with the Gong (2003), Salter et al. (2015) and Ioannidis et al. (2023) source function formulations, as shown in Figure A4. In particular, open ocean emissions from Nilsson et al. (2001) are very close to Salter et al. (2015). This comparison with usual open ocean source functions shows that the approach consisting in leveraging measurements from Nilsson et al. (2001) is fit for our purpose.

### 3.7 Implementation of the lead parameterization in the WRF-Chem model

In order to assess the regional impact of sea spray emissions from leads on aerosol concentrations, the parameterization for sea spray emissions from leads studied throughout this work is also implemented in the 3D regional atmospheric chemistry model WRF-Chem 4.3.3 (code version available at Lapere (2024)). The WRF-Chem model is commonly used for Arctic case studies and generally shows good performance in reproducing atmospheric composition and aerosols in this region, including sea spray (Marelle et al., 2017, 2021; Raut et al., 2022; Ahmed et al., 2023; Ioannidis et al., 2023). A 2-month simulation is conducted for August–September 2018. Summertime is considered because it was identified as the season when sea spray emissions from leads are the highest in Section 3.2.

### 3.7.1 WRF-Chem model setup

A sensitivity analysis, with and without sea spray from leads, is performed for the period August 1 to September 30 2018, including 17 days of spin-up from July 15 to July 31. The simulation domain at 50 km spatial resolution with 72 vertical levels up to 50 hPa, comprising the Arctic Ocean, is described in Figure 10.

The sea spray source function used in the WRF-Chem simulations is described in Ioannidis et al. (2023). This source function combines the size distribution from Gong et al. (1997), the wind dependence of the flux from Salisbury et al. (2014) and the SST dependence from Jaeglé et al. (2011). It was shown to be a suitable option for open ocean Arctic sea spray emissions (Ioannidis et al., 2023). The $R_{\mathrm{Nilsson}}$ ratio is then applied on this source function to provide a middle range estimate of the

impact of leads on sea spray. The lead fraction is extracted from the neXtSIM product and is taken as the fraction of open water wherever sea ice concentration is greater than 80%. Sea ice concentration is also taken from neXtSIM in all simulations.

Sea spray emissions in WRF-Chem consist of sodium, chloride, sulfate and organics. The sulfate fraction is computed based on the measurements from Calhoun et al. (1991), and the Vignati et al. (2010) organic fraction is used for primary marine organic emissions (which are attributed to the organic carbon (OC) WRF-Chem species in this model version). Blowing snow emissions of sea salt aerosol are also activated, following the implementation from Marelle et al. (2021). The aerosol scheme is the Model for Simulating Aerosol Interactions and Chemistry (MOSAIC) with 4 bins (Zaveri et al., 2008) and the

initial and boundary conditions for atmospheric composition are taken from the Community Earth System Model 2.2 with the Community Atmosphere Model with Chemistry (CESM2.2: CAM-Chem) (Tilmes et al., 2022). Further details of the modeling setup, including boundary and initial conditions and selected parameterizations for meteorology and chemistry can be found in Table A1.

    In order to derive the contribution of leads to sea spray aerosols, two simulations are conducted. A first simulation, referred

to as BASE, is conducted with no sea spray emissions from the marginal ice zone nor pack ice, i.e. sea spray is disabled as soon as sea ice concentration is greater than 15%, consistent with the definition of the marginal ice zone given in Vichi (2022). The second simulation activates the lead parameterization described in Section 2.1.1, applied where lead fractions are strictly positive, i.e. where sea ice concentration is greater than 80%. This case is referred to as LEADS. In order to isolate the contribution from leads, in both cases no sea spray is considered in the marginal ice zone, between 15% and 80% sea ice

concentration, and open ocean is considered where sea ice concentration is below 15%. The difference between LEADS and BASE yields an estimate of the aerosol concentrations attributable to sea spray emissions from leads.

    The WRF-Chem simulations are evaluated against measurements (Na$^+$, Cl$^-$ and OC aerosols) from the Zeppelin observatory (Ny-Ålesund, Svalbard), accessed from EBAS (Norwegian Institute for Air Research, 2024). This evaluation is presented in Figure A5 and shows a good performance of the model for all three species considered. Figure 7 also shows that the

magnitude of monthly averaged Na$^+$ concentration from the WRF-Chem simulations compare well with observations at Alert and Utqiaġvik.

### 3.7.2    Impact of leads on sea salt aerosol in WRF-Chem

The location of leads in WRF-Chem as extracted from neXtSIM is shown in Figure 10a. For the simulation period, leads are mostly found near the North Pole and in the Russian sector of the high Arctic. The lead fractions of up to 10% on average induce

a change between the LEADS and BASE simulations in Na aerosol surface mass mixing ratio (Figure 10b) by +2 ng kg$^{-1}$ on average in the high Arctic (black contour in Figure 10b). Locally, this change can be as high as +22 ng kg$^{-1}$. These absolute changes correspond to relative changes of 12% on average up to more than 30% of Na aerosol surface mass mixing ratio in the areas of high lead fraction in the high Arctic (Figure A6a). For grid points where lead fractions are more than 5% on average, sea salt emissions from leads account for an average increase of Na aerosol surface mass mixing ratio by +3 ng kg$^{-1}$,

corresponding to 18%. Therefore, sea salt emitted from leads accounts for a significant fraction of surface Na aerosol in the Arctic.

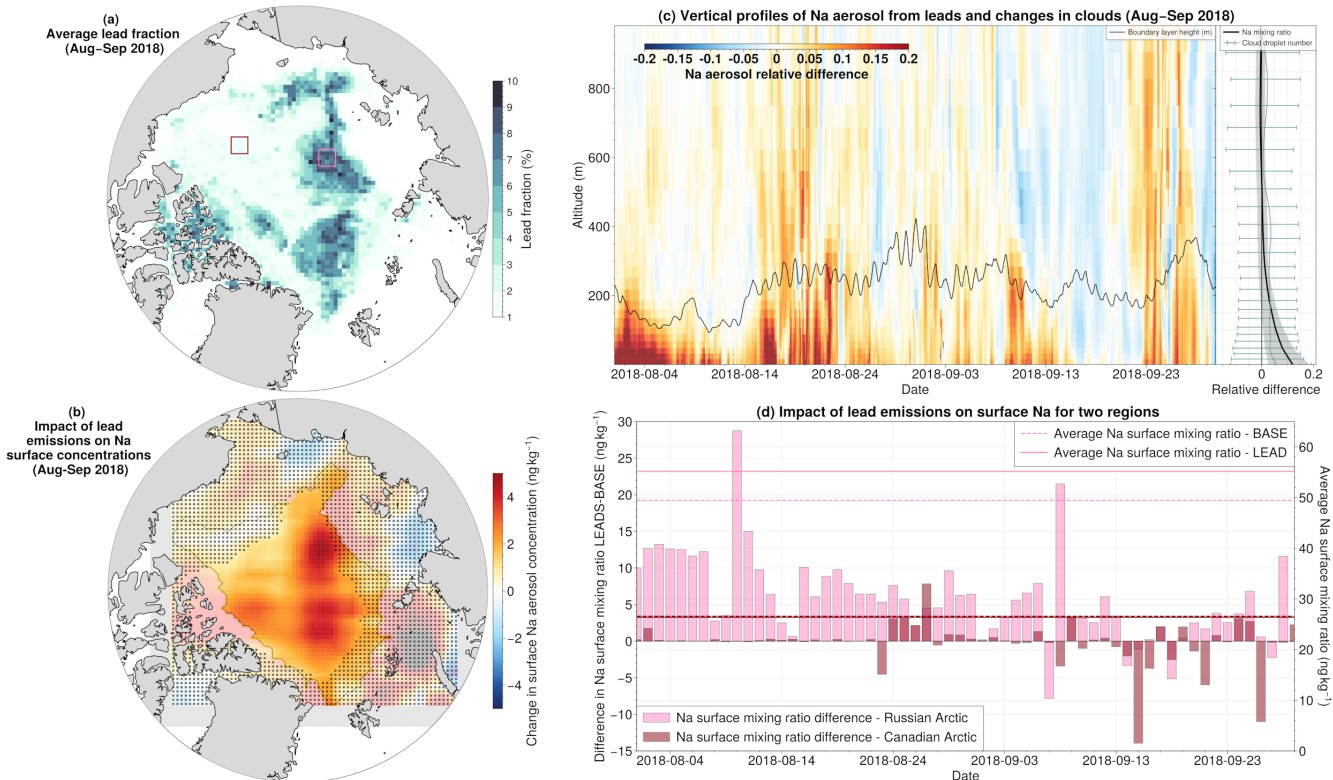

**Figure 10. Impact of sea spray emissions from leads in WRF-Chem.** (a) Average lead fraction over the simulation period extracted from neXtSIM. (b) Difference in surface Na mixing ratio between the LEADS and BASE runs. Average over August–September 2018. Stippling indicates grid cells where the change is less than 5% of the BASE run mixing ratio. Grey band indicates the boundaries of the simulation domain. (c) Vertical profiles of relative change between LEADS and BASE averaged over the high Arctic mask in panel (b). Left is the time evolution of the Na mixing ratio vertical profile, right is the average over the time period for Na (black) and cloud droplet number concentration (blue). Solid lines indicate the mean, error bars and shades indicate the 25$^{th}$ and 75$^{th}$ percentiles. (d) Change in daily Na surface mixing ratio in a Canadian Arctic region (red box in panel (a) - dark red lines/bars) and in a Russian Arctic region (pink box in panel (a) - pink lines/bars). Bars indicate the difference between LEADS and BASE, lines indicate the average mixing ratio over the period in each simulation.

The Na aerosol mass mixing ratio is affected by emissions from leads not only at the surface but throughout the mixing layer, with an impact decreasing with altitude, down to around 2% near the top of the PBL, in the high Arctic on average (Figure 10c). These changes include regional differences, as illustrated in Figure 10d. The Canadian Arctic has very low lead fraction during the simulation period (red box in Figure 10a), which results in small changes in Na mixing ratio over the period (less than 1% on the average mixing ratio), although there can be some important changes from day to day (up to 35%) due to the perturbation of the system induced by the additional emissions at the scale of the simulation domain. In the Russian Arctic where lead fractions are higher (pink box in Figure 10a), the average Na surface mixing ratio increases by 12% in the

LEADS case, with daily changes up to 49%. Therefore, sea salt from leads affects the Arctic in a heterogeneous manner, and can account for some of the spatial variability of atmospheric sea salt in the Arctic.

Because of the coupling between the meteorology and the atmospheric composition in WRF-Chem, numerical noise in the sensitivity analysis arises from changes in wind speed and boundary layer height over open ocean areas (Figure A6b,c), which result in changes in Na mixing ratio even over open sea areas. However, these changes outside the sea ice region are smaller than 5%, as indicated by the stippling in Figure 10b and Figure A6a, whereas changes due to sea salt emissions from leads contribute more than 5% over the pack ice.

The changes in cloud cover are also noisy due to the perturbation induced by the changes in Na mixing ratio. Although the average change in cloud droplet number is positive throughout the high Arctic for all vertical levels (around 3% increase compared to the BASE simulation), and despite the distribution being skewed towards positive values, the 25th percentile is on average negative (error bars in the rightmost panel in Figure 10c), and the confidence interval at the 95% level of the mean difference according to a T-test contains zero for all vertical levels. Furthermore, although marine organic emissions are included in this version of WRF-Chem, following the methodology described in Section 2.1.2, their role as INP has yet to be parameterized. Therefore, as of this version of WRF-Chem, the impact of marine organics from leads on ice clouds is not studied.

## 4 Conclusions

Based on aerosol fluxes measured in the high Arctic by Nilsson et al. (2001) and sea ice products from numerical models and satellite detection, we propose a parameterization for sea spray emissions from leads in the Arctic Ocean. Using this parameterization under different assumptions (sea ice data, source function, confidence interval of the measurements), we derive upper and lower bounds for the contribution of leads to sea salt aerosol in the high Arctic, and investigate its seasonality. Leads contribute between 0.3% and 9.8% of the annual sea salt particle number flux emitted locally in the pack ice (i.e. leads + blowing snow), but 30% to 85% of the sea salt aerosol mass flux. The asymmetry between number and mass is connected to the parameterizations of size distributions used in this work, which both for blowing snow and leads are still relatively uncertain and need further observational data. The total annual emitted mass, up to $2.1\,\mathrm{Tg\,yr^{-1}}$, is of the same magnitude as the mass of sea salt transported from the open ocean into the high Arctic in reanalysis data, revealing the critical importance of leads for the aerosol budget in the Arctic. The seasonality of sea spray from leads is found to be anti-phased with blowing snow, with maximum fluxes in summertime, while blowing snow sea salt aerosol is generated in larger quantities in the winter. We conclude that sea ice sourced sea salt aerosols are needed in models to reproduce observed seasonal variations at high latitude.

Furthermore, based on the Vignati et al. (2010) parameterization for sea spray OF, we show that is it possible to reasonably model the organic content of lead-generated sea spray, using oceanic chlorophyll-a concentration from reanalysis data. Our results agree decently with observed organic fractions of 20%–50% in the high Arctic, and show greater organic fraction in the fall, as opposed to the open ocean where the organic fraction peaks in the spring. This point is critical for modeling clouds, including cloud phase in the Arctic and further highlights the importance of modeling sea spray from leads.

The parameterization developed in this work is also implemented in the WRF-Chem model, and a sensitivity test is conducted for two months in summertime. The simulations show that including sea spray from leads increases sea salt mass mixing ratio by around 12% in the high Arctic for August–September 2018, with regional differences. The important emission fluxes and contribution to concentrations found in this work, and the significant organic fraction in sea spray from leads, suggest that sea spray from leads could have important impacts on cloud, which have yet to be better estimated.

Currently, the largest uncertainty for constraining the sea spray emission flux from leads comes from the confidence interval of the observations used to build the parameterization. In particular, not only the magnitude but also the sign of the variation of emissions from leads with wind speed is not clearly established, depending on the chosen $R_{\mathrm{Nilsson}}$, as illustrated in Figure 1a, with increasing $R_{\mathrm{Nilsson_{max}}}$ but decreasing $R_{\mathrm{Nilsson_{min}}}$ as a function of wind speed. This result should motivate future measurement campaigns with designs such that the observed data can readily inform models. In parallel, modeling experiments using the parameterization proposed in this work should be conducted on longer time scales to further estimate the impact of leads on the seasonality of aerosol populations throughout the Arctic atmosphere, including their role in the Arctic climate. Furthermore, because of the choice of lead detection product and because we ignored emissions from the MIZ throughout this work, the estimates of sea spray fluxes presented here are likely lower bounds, which further illustrates the importance of leads for the Arctic aerosol budget. Additionally, although developed from an Arctic perspective, the parameterization could be tested for lead emissions in Antarctic sea ice as a first estimate.

Finally, the parameterization proposed in this work can be leveraged to study the relative importance of open ocean/transported sea spray particles versus particles from open water areas in sea ice regions, in the context of changes in sea ice and extent of the MIZ. However, as highlighted here, a better understanding of aerosol sources in the MIZ and associated parameterizations are still missing to better comprehend Arctic aerosols.

*Code and data availability.* The Python Jupyter Notebooks in which the parameterizations have been implemented are publicly available on Zenodo at https://doi.org/10.5281/zenodo.10782399 (Lapere, 2024). The WRF-Chem model including our lead emission parameterization is available at https://doi.org/10.5281/zenodo.10782399 (Lapere, 2024)

*Author contributions.* Conceptualization: RL, JLT, LM, PR, GS, CM. Formal Analysis: RL, JLT, PR. Data curation: RL, LB. Investigation, Methodology: RL, JLT. Software: RL, LM. Visualization: RL. Funding Acquisition: JLT. Writing – original draft preparation: RL, JLT. Writing – review & editing: LM, PR, LB, GS, CM.

*Competing interests.* The authors declare that they have no conflict of interest.

*Acknowledgements.* This project has received funding from the European Union's Horizon 2020 research and innovation programme under Grant Agreement No 101003826 via project CRiceS (Climate Relevant interactions and feedbacks: the key role of sea ice and Snow in the polar and global climate system) and from the Horizon Europe programme under Grant Agreement No 101137680 via project CERTAINTY (Cloud-aERosol inTeractions & their impActs IN The earth sYstem). The WRF-Chem simulations were performed using HPC resources from GENCI–IDRIS (Grant A0150107141). We thank J.-C. Raut, D. Voisin, and H. Angot for the fruitful discussions. Finally we would like the thank the three reviewers for their valuable insights and comments, and the editor for managing this manuscript.

# Appendix A: Appendix

## A1 Formulations of sea spray source functions

The Gong (2003) source function describes the wind-dependent sea spray number flux density following Equation A1.

$$\frac{dF}{dr} = 1.373u^{3.41}r^{-4.7(1+30r)^{-0.017r^{-1.44}}}(1+0.057r^{3.45})10^{1.607\exp(-((0.433-\log r)/0.433)^2)} \tag{A1}$$

with $u$ the 10 m wind speed, and $r$ the particle radius.

The Salter et al. (2015) source function, which includes a dependency on SST, expresses the sea spray number flux density based on air entrainment, as described in Equation A2, using three log-normal modes defined by 6 parameters each ($D_{0i}$ the modal diameters, $\sigma_i$ the geometric standard deviations, and $A_i$, $B_i$, $C_i$, $D_i$ a set of fitting parameters).

$$\frac{dF}{d\log D} = 10^{-8}u^{3.74}\sum_{i=1}^{3}\frac{(A_iT^3+B_iT^2+C_iT+D_i)}{\sqrt{2\pi}\log\sigma_i}\exp(-\frac{(\log D-\log D_{0i})^2}{2\log\sigma_i^2}) \tag{A2}$$

with $u$ the 10 m wind speed, T the SST in °C, and $D$ the particle diameter.

The Ioannidis et al. (2023) source function combines the Gong et al. (1997) size distribution with the whitecap fraction from Salisbury et al. (2014) and the SST dependence from Jaeglé et al. (2011), and includes a correction for smaller particles based on observations from O'Dowd et al. (1997). Its formulation is described in Equation A3.

$$\begin{cases} SST_{fac} = 0.3+0.1T-0.0076T^2+0.00021T^3 \\ \frac{dF}{dr} = 4.60e^{-5}u^{2.26}3.6e^5r^{-3}(1+0.057r^{1.05})10^{1.607\exp(-((0.380-\log r)/0.650)^2)}SST_{fac} \\ \frac{dF}{dr} = \frac{dF}{dr}\exp(-0.5(\log(r/0.1)/\log(1.9))^2) \qquad \text{for r<0.1} \end{cases} \tag{A3}$$

with $u$ the 10 m wind speed, T the SST in °C, and $r$ the particle radius.

## A2 Sensitivity of the parameterization

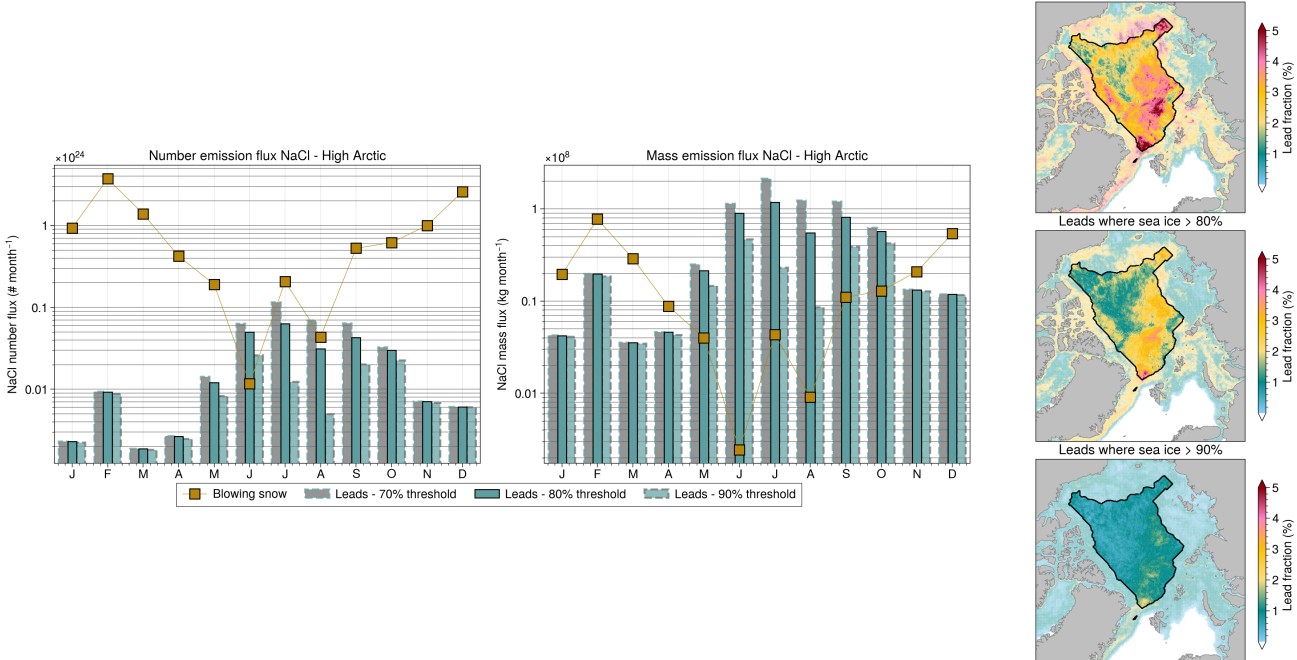

**Figure A1. Sensitivity to lead threshold definition.** Sea salt aerosol emissions from leads in the high Arctic using 80% (darker blue bar, solid line), 70% (gray bar, dashed line) or 90% (lighter blue bar, dashed line) sea ice concentration threshold. Brown markers show the particle number flux from blowing snow as a reference. The lead fluxes are the average of the three source functions using $R_{\mathrm{Nilsson}}$ ratio. The sea ice concentration is from neXtSIM. Left panel is the particle number flux, middle panel is the particle mass flux. The right panel shows the annual average lead fraction obtained with each threshold.

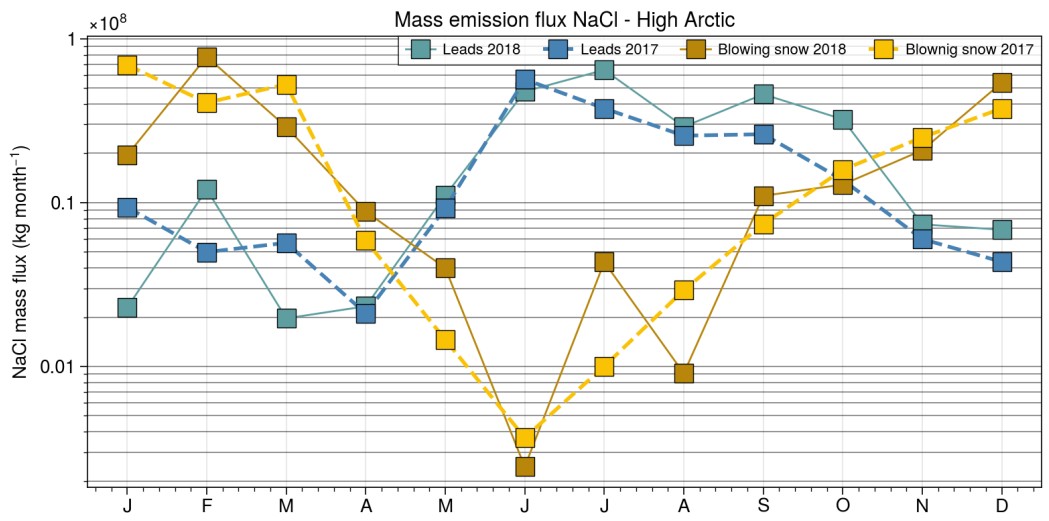

**Figure A2. Year-to-year variability.** Sea salt aerosol mass flux from blowing snow (yellow) and leads (blue) in the high Arctic for 2018 (darker colors, solid lines) and 2017 (lighter colors, dashed lines). The source function used for lead emissions is Gong (2003) with $R_{\mathrm{Nilsson}}$ ratio. The sea ice concentration is from neXtSIM.

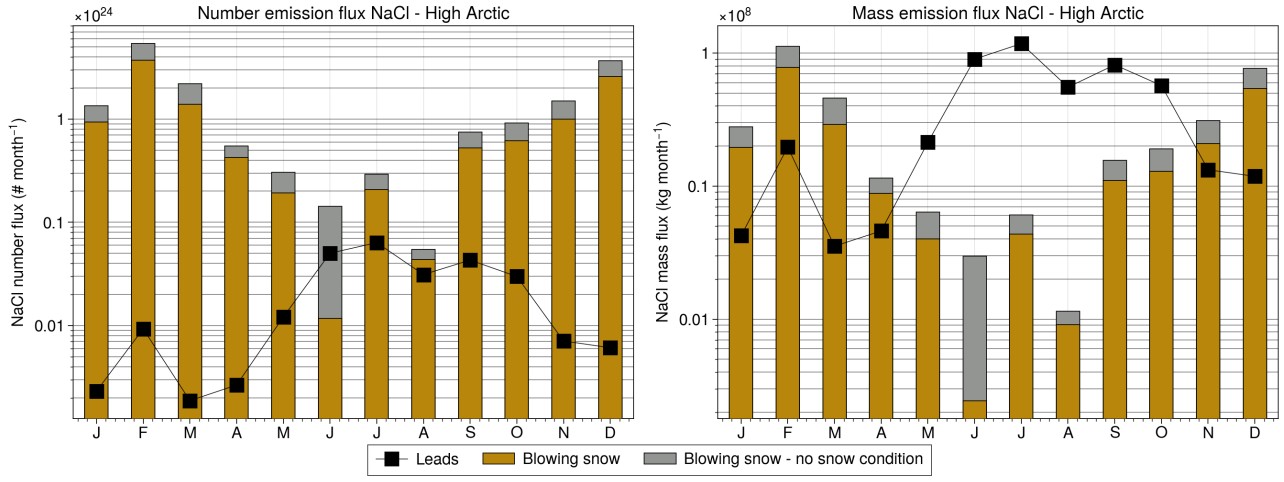

**Figure A3. Sensitivity of blowing snow emissions.** Sea salt aerosol emissions from blowing snow in the high Arctic with (brown bar) and without (gray bar) the condition on snow thickness variations described in Section 2.1.3. Black squares indicate the emissions from leads for reference (average of the three source functions using $R_{\mathrm{Nilsson}}$ ratio). Left panel is for particle number flux, right panel is for particle mass flux.

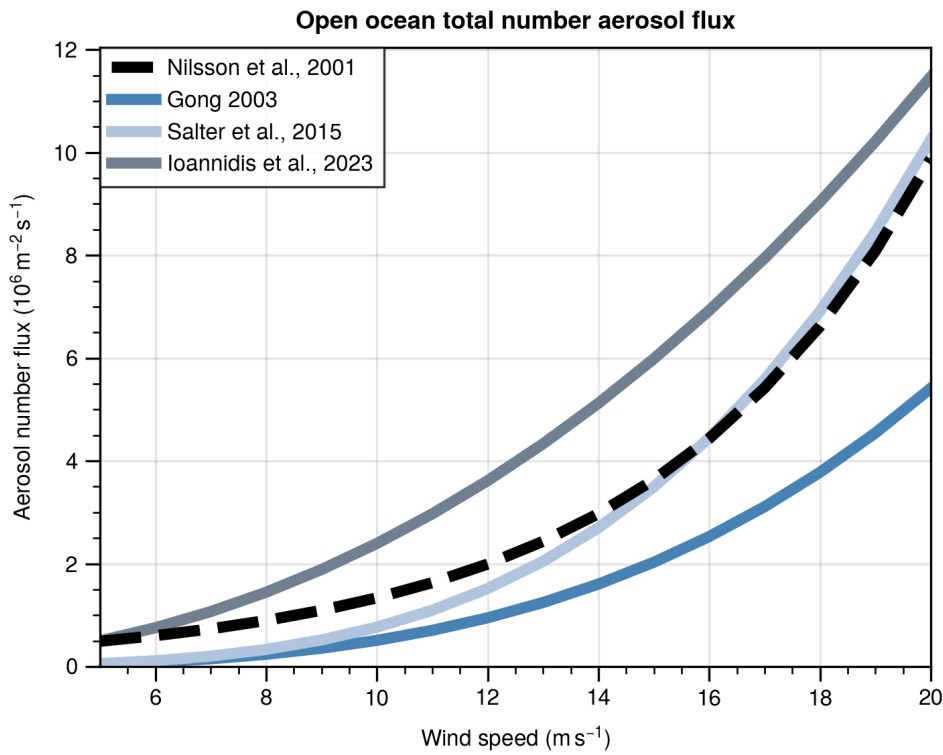

**Figure A4. Nilsson et al. (2001) fluxes compared to usual source functions.** Total aerosol number flux as a function of wind speed from the Nilsson et al. (2001) measurements (dashed black line - best fit coefficients) and the Gong (2003) (darker blue line) and Salter et al. (2015) (lighter blue line) source functions. The size distribution used for the source functions is as described in Section 2.1.1.

## A3 WRF-Chem setup and additional results

| Physics and Meteorology | Model Option |
| --- | --- |
| Planetary boundary layer / Surface layer | MYNN 2.5 level TKE scheme / MYNN (Nakanishi and Niino, 2009) |
| Surface layer | Noah LSM (Tewari et al., 2004) |
| Microphysics | Morrison (Morrison et al., 2009) |
| SW/LW radiation | RRTMG (Iacono et al., 2008) |
| Cumulus | Grell-3 (Grell and Dévényi, 2002) |
| Meteorology IC and BC | NCEP FNL (National Centers for Environmental Prediction, 2000) |
| **Aerosol and chemistry** | **Model Option** |
| Gas-phase chemistry | MOZART incl. aqueous phase chemistry (Emmons et al., 2010) |
| Aerosols | MOSAIC-4bin (Zaveri et al., 2008) |
| Chemical IC and BC | CESM2.2: CAM-Chem (Tilmes et al., 2022) |
| **Emissions** | **Model Option** |
| Sea spray | Ioannidis (Ioannidis et al., 2023), with organics from (Vignati et al., 2010) |
| Anthropogenic | ECLIPSE v6b (Klimont et al., 2017) |
| Fire | FINNv1.5 (Wiedinmyer et al., 2014) |
| Biogenic | MEGAN (Guenther et al., 2012) |
| DMS | Lana2011 (Lana et al., 2011) |

**Table A1.** WRF-Chem model setup

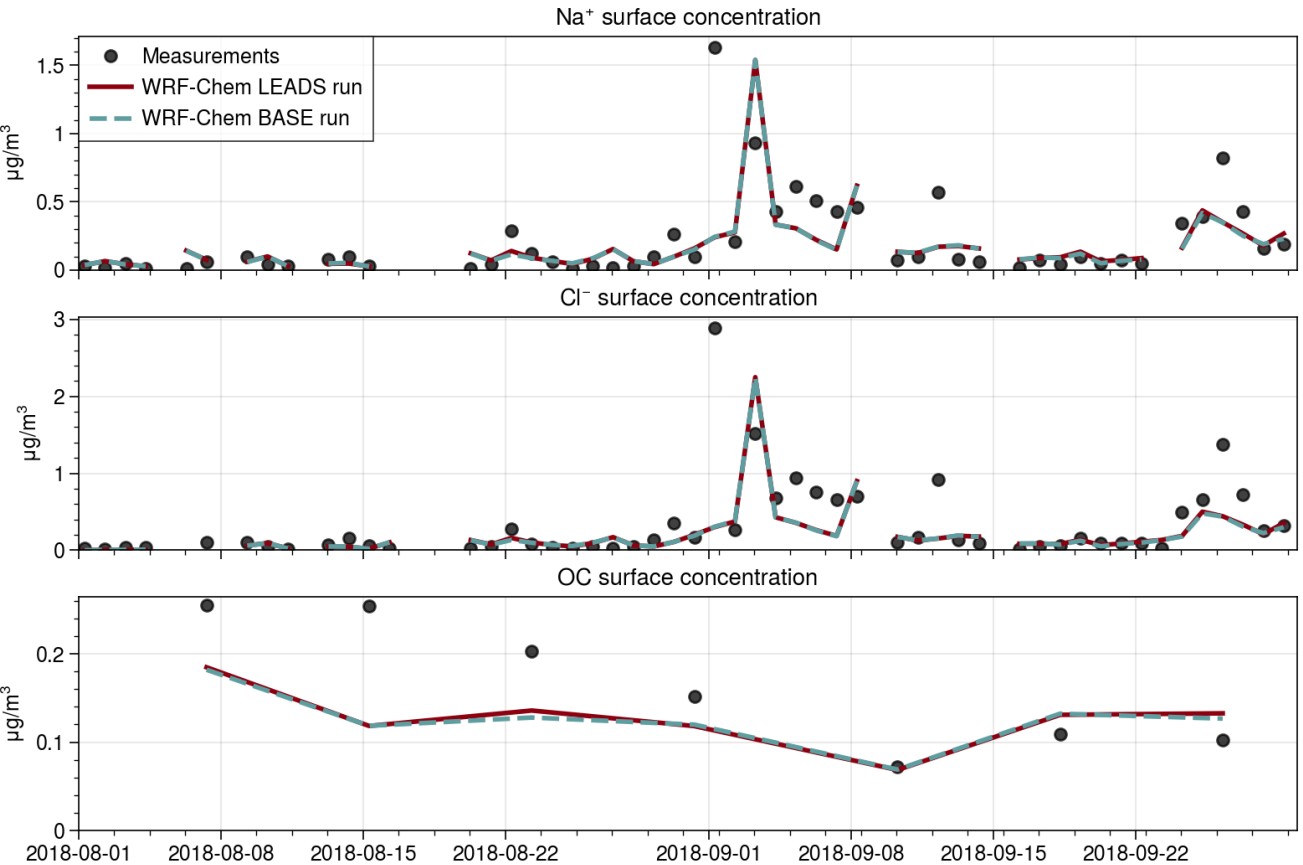

**Figure A5. WRF-Chem evaluation.** Surface concentration of $Na^+$, $Cl^-$ and OC at Zeppelin station, Svalbard, in WRF-Chem runs (red is with leads, blue is without leads) and measured (dots). Observations are total suspended particles.

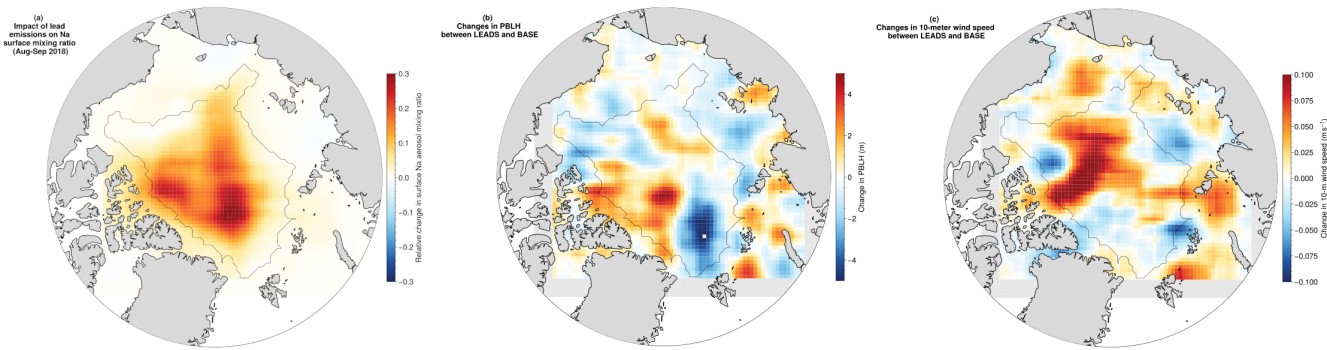

**Figure A6. WRF-Chem sensitivity.** (a) Relative difference in Na surface mixing ratio. (b) Difference in PBL height. (c) Difference in 10-meter hourly maximum wind speed. All panels show average differences between LEADS and BASE simulations for August–September 2018.

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
