# Peer review of "Modeling the contribution of leads to sea spray aerosol in the high Arctic"

_EGUsphere, 2024_

## Referee Comment (RC1)

Review for "Modeling the contribution of leads to sea spray aerosol in the high Arctic", by Lapere et al.

This study investigates the contribution of leads on sea spray emissions in the high Arctic. The authors first are using a parametrization based on a field work study and they are incorporating it with current state of the art sea spray functions to estimate sea salt emissions from leads. Then they are applying their parametrisation to Weather Research and Forecasting (WRF) model coupled with Chemistry for a case study. This is an innovative work as such parametrizations are missing from regional transport models. However, the manuscript can be further improved. I therefore recommend this manuscript to be published in ACP once the minor comments below will be addressed.

First, I have some general comments. Please address:
Move the figures closer to where they are mentioned in the text, otherwise it is difficult to read your manuscript and look for the figures.
Also, throughout the text you either discuss Gong et al. 2003 or both Gong et al. 2003 and Salter et al. 2015. Please be consistent and always discuss both.
You are using abbreviations throughout the text that they have not been defined. Please define.

Minor comments:
Line 16: Use abbreviation for sea spray aerosols (e.g. SSA)
Line 23: Use abbreviation for sea spray emissions (e.g. SS emissions)
Lines 23-29: You are discussing about the main source of sea spray aerosols, as well as for Arctic related sources. You do not mention frost flowers as a potential source. Please mentioned them in your discussion in the introduction to be complete as there are papers looking into that.
Line 40: Define sodium as $Na^+$ and use throughout the text.
Line 81: Define abbreviation "WRF-Chem".
Lines 96-104: You are mentioning that you are using Gong et al. 2003 and Salter et al. 2015 but you are not providing a satisfactory explanation. Explain better why you chose these two since there are plenty more source functions. Barthel et al. 2009 for example, tested more than five source functions. It would be interesting to see results for one-two more source functions. Also please add the source functions (e.g. Gong et al. 2003, Salter et al. 2015 or if you use more) in your manuscript (with and without Nilsson correction).
Line 117: Explain why you are choosing Vignati et al. 2010?
Line 125: You are using Vignati et al. 2010 in your WRF-Chem simulations. However, Archer-Nicholls et al., 2014 already included a source function for marine organics following Fuentes et al. 2010 which was later corrected for Arctic as shown in Ioannidis et al. 2023. Please mention why did you decide to change the source function and how does Vignati compare with original Fuentes 2010 and/or Ioannidis et al. 2023?
Line 147: Mention what is HYCOM-CICE and TOPAZ
Line 156: Define the abbreviation "NEMO"
Line 157: Define the abbreviation "NANUK" and move the definition of neXtSIM in the previous line.
Line 168: Define the abbreviation "MODIS".
Line 194: Define the abbreviation "ERA5".

Line 195: Define the abbreviation "MERRA-2".

Line 197: Use Na$^+$ for sodium and define once first appearing in the text.

Line 215 and Figure 2: You say you are focusing on the high Arctic. However, have you considered to use the radar data for sea ice at Utqiaġvik, Alaska to address the uncertainties in the different products you are using near the coastlines and even correct your parametrization for coastline regions. May et al. 2016 and Kirpes et al. 2019 are discussing radar data so they might be available.

Figure 3 and Figure 4: Show results for Salter et al. 2015 not only Gong et al. 2003. Or at least justify why you are showing results only for Gong et al. 2003. In Figures 5, 6 and 7 you are showing results for both so it's really confusing.

Figure 7: Replace sodium with Na$^+$. Also mention if observations are sub-micron, super-micron, PM2.5 or PM10.

Lines 269-272: Add a reference.

Lines 290-291: Provide more information about the observations, are they sub-micron? Super-micron? Unclear.

Lines 307-310: Compare/discuss why you didn't choose Fuentes et al. 2010 or Ioannidis et al. 2023.

Lines 325-327: Please re-read Kirpes et al. 2019. They did not measure atmospheric concentrations. They measured elemental fractions. Also be more precise. They used ratios as an indicator of local influence or transported and they found a local influence on sea spray aerosols. Re-write this part.

Line 332 and Figure A1: Again, you only show results based on Gong et al. 2003. Please decide and show only Gong et al. 2003, and justify why, or both Gong et al. 2023 and Salter et al. 2015.

Figure 9: Compare with Fuentes et al. 2010.

Line 376: You are using WRF-Chem model which uses a different source function to estimate sea spray aerosols compared to your analysis so far. Why did you not include Ioannidis et al. 2023 source function in your analysis so far? You could add the source function in Figure 2 and in the text, as the other two (see earlier comment). It would make much more sense to include the source function you are going to use in your WRF-Chem simulations in your earlier analysis and apply there the Nilsson et al. 2001 parametrization/ratio you got and trying different sea ice products. Maybe you will get different results. Or if you already tested it then please discuss it/mention it in the text. Would be interesting to see how Ioannidis et al. 2023 compares with the rest in your analysis, since it was tested for simulations over the Arctic.

Define what SSA is in the WRF-Chem model version you are using.

Line 380: You are using NCEP FNL reanalysis data. I assume sea ice fraction is coming from there. Did you consider using NEMO-netXtSLM sea ice concentration in your WRF-Chem simulations, which gave you the best lead estimations? Can you use NEMO-netXtSLM in your simulation instead of NCEP FNL sea ice fraction? Or did you compare NCEP FNL from the three sea ice products you used to estimate potential uncertainties you might get in your model results coming from NCEP FNL?

Also, for your simulations are using NCEP FNL while for your earlier estimations you used ERA5. ERA5 and NCEP FNL have some differences. Try to explain the inconsistencies and the uncertainties you introduce to your parametrization by switching reanalysis data.

Line 370 and after: You add a new source in the model. How does this affect dry deposition in the model? How does this influence other inorganic aerosols?

Line 380: You just mention the name of the model options for your simulations in the APPENDIX. Add text where you explain briefly the most important options for your simulations (such as aerosol scheme). Add citations that WRF-Chem has been used for Arctic simulations and that can be trusted.

Line 418: Did you test how different will be your results if you use Fuentes et al. 2010 or Ioannidis et al. 2023 with adjusted Fuentes?

Figure 10: You present your parametrisation implemented in a regional transport model. How do you make sure that your parametrization is trusted? How does the model compare against observations?

 Please compare against observations. How do you know your results make sense and can be trusted? Compare against sub-micron, super-micron and PM10 $Na^+$ observations at least in a few Arctic sites. Also include comparison against $Cl^-$.

Did you estimate ratios, such as $Cl^-:Na^+$, $SO_4^{2-}:Na^+$ as Kirpes et al. 2019 or did you calculate depletion factors as Frey et al. 2020 to be able to provide more robust conclusions about your model results?

---

## Referee Comment (RC2)

This study makes use of measurements of particle emission flux from Arctic leads to produce a parameterisation of the process that is suitable for models. The work is a novel contribution to a research question that is highly relevant to the scope of ACP, since this is a key polar aerosol process that is missing in current state-of-the-art climate models. The parameterisation is tested both using inputs from reanalysis to calculate emissions, and within the Weather Research and Forecasting model with chemistry (WRF-Chem). Results are compared to measurements where possible, and the relative importance of leads as an aerosol source compared to blowing snow and transported sea salt is assessed. I recommend that this article be published following some minor comments detailed below.

Ruth Price

**General comments**

Emissions from the marginal ice zone (MIZ) are largely excluded from the parameterisation. I acknowledge the justification for doing this in section 2.2.4, and analysis of the sensitivity of the results to this choice in Figure A1 and section 3.6. However, given the importance of the MIZ in the question of how Arctic aerosol will respond to future warming, some more discussion of modelling emissions from the MIZ would be welcome in the introduction and in 3.6 or 4. Specific recommendations about this are included below.

Regarding figure 7 and the associated discussion, the potential role of transport and wet removal has been neglected as a potential source of disagreement between the conceptual modelling and the measured concentrations. Please address this in the text (see below).

There is some confusion caused by terminology in figure labels and captions, for example mixing "monthly sum" and "daily average" in the same figure or using the word "flux" both for emissions per unit area per unit time and simply for emissions per unit time. Please address these so that there is consistency throughout.

**Specific comments**

Lines 21-22: sea spray has also been shown to have a key effect on the climate impact of secondary aerosol particles in the changing Arctic climate system, by changing the condensation sink (Browse et al. 2014) but this is by no means well established because results vary across models (Gilgen et al. 2018).

Line 33-35: it is important to highlight (here or elsewhere in introduction) why sources from leads are an important factor in establishing the sensitivity of Arctic aerosol to sea ice in the present day and in the future (or other differing climate states). Eg in the warming Arctic as sea ice extent/thickness changes and the marginal ice zone becomes wider at certain times of year (Strong and Rigor 2013), emissions from leads may be relatively more or less important than emissions from open ocean or pack ice. So without including this process in models, our knowledge of the changing natural aerosol baseline in the Arctic may be incomplete. (This point is made via the inclusion of the question in line 76, but the motivation could be made stronger).

Figure 1b: since the Salter et al emission parameterisation includes an SST dependence, it's not clear whether the size distribution shown here in the figure is for a particular value of SST. Please give this information in the figure or caption.

Line 128: "…only affect the sea salt emissions, but not…" surely it will also affect the organic emissions? Perhaps clearer to say "only affect the composition of the emitted particles, but not the absolute emissions flux of sea spray".

Line 195: the important conclusion that emissions from leads/blowing snow could be as important for the Arctic aerosol budget as transport relies on the comparison with MERRA-2 data. As such, please give some information here on how MERRA-2 sea salt aerosol is modelled (emission, removal processes) - has the representation of aerosols in MERRA-2 been previously evaluated for the Arctic? In particular I think it is important to highlight any uncertainties in the calculation of sea salt transport in MERRA-2 that could impact the comparison.

Lines 202-203 and figure 2: it's not clear which underlying emissions flux is used to calculate the fluxes presented in figure 2(d-f) (although the figure caption for figure 3 states Gong (2003) is used, which suggests the same is true for figure 2). Please give this information here in the text or in the caption for figure 2, or both.

Figure 3 and 5: the y-axis label gives these values as an "average rate" in #/s, while the figure caption states "monthly sum of sea salt number emissions". As such, I struggle to interpret whether the values represent total number of emitted particles calculated over each month (with units of particle number), or an average emission rate of particles for the month. Moreover, the y-axis label refers to a "daily average" which conflicts with the monthly aggregation here. I interpret this to mean that the aggregation is sampled from daily mean flux values, but I don't think it's necessary to write that here. The daily meaning is described in the text (lines 120, 163, 195) and I think that's sufficient. Alternatively if I have interpreted this wrong, please clarify the meaning of "daily average" as used here.

Figure 4: similar to previous comment on figures 3 and 5, same note here about the use of the phrase "daily average" to describe seasonal means.

Figures 7 and 9: values in maps are given units of [mass per unit time], whereas I would expect [mass per unit area per unit time] for a mass flux, and this would seem to be more consistent with your description of normalising the flux values by PBLH (in fig 7). Please double check these units.

Figure 7 and line 294-296: comparing emissions data to concentration measurements: "…which signals that our parameterization could be improved despite already showing better agreement…" The conceptual model is of local emissions only (with some accounting for vertical mixing via the use of PBLH) whereas measured concentrations have also been influenced by removal and transport. As such this conceptual model of the "usual approach" is not totally consistent with what a climate model would produce when including explicit representations of the vertical mixing, transport, and removal processes. Given the fluctuations in transport and removal throughout the year at these locations, it seems possible that the wintertime underestimation and summertime overestimation by "usual approach" could be influenced by significant contributions from long-range transport and wet deposition (respectively). This should be emphasised in the text to highlight the uncertainties in modelling polar sea salt aerosol other than just these missing sources. Also, it may be informative to calculate the sodium concentrations for these locations from the WRF-Chem output (where transport and removal are included) and add those values here as datapoints for the months available.

Figure A1: it would be instructive to see maps of the extent of the revised "high Arctic" areas using the different sea ice thresholds, to understand how the choice of threshold influences the spatial area under consideration. Moreover, one interesting application of the parameterisation introduced

in this study will be to study the effect of sea ice changes on the future Arctic aerosol budget – specifically, the changing extent of the MIZ relative to total ice extent and how this may change the relative importance of open ocean/transported sea spray particles vs particles from open water areas in sea ice regions. It would be good to add some discussion of this aspect of the research either here or in the conclusion section. Specifically, it is important to emphasise the need for a better understanding of aerosol sources in the MIZ (including sea spray) and a need for this to be translated into climate models via parameterisations such as the one presented in this study.

Line 444-445: it's shown in Figure 1 that it is not just the magnitude of the emissions that is uncertain, but even the sign of the response with increasing wind speed is unconstrained. This is an interesting result that could be stressed here.

**Technical corrections**

Line 92: "the total particle number aerosol flux over the open Arctic Ocean was…" open ocean measurements reported in Nilsson are from the Barents Sea region further south rather than the Arctic Ocean itself. Please double check this and amend your statement here.

Figure 6: y-label is not very intuitive, suggest to change to something like "Total sea salt source / Gg month-1"

Figure 8: please explain exactly which areas are included in "open ocean", i.e. what's the southerly extent.

Figure 9: the map of Utqiagvik here uses a different projection from that in figure 7 - suggest to use a consistent projection for both.

Line 427: "This difference mass and number" should read "difference in"?

Lines 638-end: should the references by Ólason et al. come under "O"?

---

## Author Comment (AC1)

**Response to the reviewers for manuscript egusphere-2024-1271**
Modeling the contribution of leads to sea spray aerosol in the high Arctic
Lapere et al.

Dear Editor and referees,

We would like to thank the Editor and the three referees for their reviews and suggested revisions of our work. All of the comments were addressed in the response to reviews and appropriate revisions are included in the new version of the manuscript. Please note that answers are in the boxes after each comment.

Summary of the main issues that have been addressed:

1. Further justification and discussion regarding some of the modeling choices made (sea spray source function, marine organic fraction) have been added.
2. The source function used in the WRF-Chem simulations is now also included in all the conceptual model calculations.

Please note: Line and figure numbers in the responses refer to the initial manuscript submission to be consistent with the reviewers' comments. The reviews are in black, the responses are in blue.

Best regards,
Rémy Lapere
9 September 2024

**Reviewer #1**

This study investigates the contribution of leads on sea spray emissions in the high Arctic. The authors first are using a parametrization based on a field work study and they are incorporating it with current state of the art sea spray functions to estimate sea salt emissions from leads. Then they are applying their parametrisation to Weather Research and Forecasting (WRF) model coupled with Chemistry for a case study. This is an innovative work as such parametrizations are missing from regional transport models. However, the manuscript can be further improved. I therefore recommend this manuscript to be published in ACP once the minor comments below will be addressed.

*General comments:*

1. Move the figures closer to where they are mentioned in the text, otherwise it is difficult to read your manuscript and look for the figures.

> We tried to put the figures as close as possible to the corresponding text while keeping them with a reasonably big enough size. Figures 7-8-9 which were a little far from the text have now been moved to make the manuscript easier to read. In any case, we will make sure that the figures appear where they should in the finalized published manuscript.

2. Also, throughout the text you either discuss Gong et al. 2003 or both Gong et al. 2003 and Salter et al. 2015. Please be consistent and always discuss both.

> In Section 3.1, the focus is on the sensitivity of the flux to the choice of surface information for determining leads. Therefore, in order to isolate this parameter, we only considered one source function. Also, because this Section investigates average fluxes, both the spatial patterns in Figure 2 and relative variations in Figure 3 are very similar, irrespective of the source function considered. However, in order to be more consistent, all parts that were considering only one source function now show the average of all three (see comment #28) source functions instead. Figures and associated paragraphs have been adapted accordingly.
>
> This approach is made clearer with the addition of the following sentence line 203:
> *"Throughout this section, the fluxes presented are the average of the three source functions, in order to isolate the impact of the choice of the sea ice data set, all other things remaining equal. Comparisons between the source functions are presented from Section 3.2 onwards."*

3. You are using abbreviations throughout the text that they have not been defined. Please define.

> The missing abbreviation definitions have been included as per the reviewer's suggestion.

*Minor comments:*

4. Line 16: Use abbreviation for sea spray aerosols (e.g. SSA)

> In this work we sometimes present results in terms of sea spray (inorganic sea salt + marine organics), but also sometimes in terms of sea salt only when the organic fraction is not considered. This is why we did not use the abbreviation "SSA" since it could be confusing between sea spray and sea salt.

5. Line 23: Use abbreviation for sea spray emissions (e.g. SS emissions)

> For the same reason as in comment #4, we prefer not to abbreviate sea spray / sea salt to avoid confusion.

6. Lines 23-29: You are discussing about the main source of sea spray aerosols, as well as for Arctic related sources. You do not mention frost flowers as a potential source. Please mentioned them in your discussion in the introduction to be complete as there are papers looking into that.

> The following is now included at line 27.
>
> *"Studies have suggested frost flowers as another potential sea ice source of sea salt aerosol in Arctic coastal regions (Domine et al., 2004; Xu et al., 2016), although the extent to which they contribute is still uncertain (Huang et al., 2018; Kirpes et al., 2019)."*

7. Line 40: Define sodium as Na+ and use throughout the text.

> The proposed modification has been made.

8. Line 81: Define abbreviation "WRF-Chem".

> The proposed modification has been made.

9. Lines 96-104: You are mentioning that you are using Gong et al. 2003 and Salter et al. 2015 but you are not providing a satisfactory explanation. Explain better why you chose these two since there are plenty more source functions. Barthel et al. 2009 for example, tested more than five source functions. It would be interesting to see results for one-two more source functions. Also please add the source functions (e.g. Gong et al. 2003,

Salter et al. 2015 or if you use more) in your manuscript (with and without Nilsson correction).

> The selection of the three source functions used in this work has been made in order to cover different families of source functions which are representative of what most models use (one classical whitecap fraction approach widely used in regional and global models, one that includes an SST dependency in addition and one that is based on entrainment factors rather than whitecap). Although we understand the interest of looking at as many source functions as possible, we cannot test more source functions without making the figures and analysis too dense. This is why we provide all the code used for this manuscript to the community, in order for anyone interested in testing a source function to be able to do so.
>
> To explain the choice of these three source functions (instead of two, see comment #28), the following text has been added (line 104):
> *"By using these three functions (which formulations are described in Section A1), we can compare an approach commonly used in global models (Lapere et al., 2023) with more Arctic-adapted ones, including with a non-whitecap-based approach. The interested reader is invited to use the code of the conceptual model provided with this study to test any other source function."*
>
> As requested, the description of the source function formulations has also been added in Appendix.

10. Line 117: Explain why you are choosing Vignati et al. 2010?

> The following is now included at line 124:
> *"We tested several parameterizations for sea spray OF which connect chlorophyll-a to OF with a simple relationship, including Vignati et al. (2010); Rinaldi et al. (2013); Gantt et al. (2015) (results not shown here). We found that Vignati et al. (2010) yielded the most realistic values, which for the Arctic compared to measurements from Leck et al. (2002) and Kirpes et al. (2019) (see Section 3.5). For consistency, the OF parameterization is used both in the conceptual model (Sections 3.1 through 3.5) and in WRF-Chem (Section 3.7) to estimate the marine organic aerosol emissions."*

11. Line 125: You are using Vignati et al. 2010 in your WRF-Chem simulations. However, Archer-Nicholls et al., 2014 already included a source function for marine organics following Fuentes et al. 2010 which was later corrected for Arctic as shown in Ioannidis et al. 2023. Please mention why did you decide to change the source function and how does Vignati compare with original Fuentes 2010 and/or Ioannidis et al. 2023?

> The goal of this study is to develop a parameterization of emissions from leads that is usable in global and Earth system models, with testing in a conceptual model and then in WRF-Chem as a first step towards this. The parameterization of marine organics used in

Fuentes et al. (2010) and Ioannidis et al. (2023) depends on seawater diatomaceous bioexudate organic carbon concentration, which is not readily available in global models or from satellite retrievals. Therefore, such an approach cannot be used in the near future for this application. Thus, we use instead the Vignati et al. (2010) marine organic fraction, which depends on the more widely available chlorophyll-a surface concentrations. The justification for this choice is now explained in the manuscript at the end of Section 2.1.2:

*"We acknowledge that there are limitations associated with parameterizing organic fraction using chlorophyll-a. However the availability of chlorophyll-a products makes it more likely that these parameterizations can be adapted and used within climate and Earth system models. While it is clear that other seawater characteristics than chlorophyll-a (such as organic carbon or glucose concentration) can better account for the OF of sea spray (Fuentes et al., 2010; Quinn et al., 2014; Rocchi et al., 2024), their lack of general availability in satellite or model-based products limits our ability to recommend these now for use by large scale models."*

Furthermore, the implementation of the Fuentes et al. (2010) parameterization in WRF-Chem relies on domain-wide parameters that are not available temporally or spatially resolved, such as a fixed pre-computed size-resolved organic fraction, or the seawater OC<0.2μm concentration. In addition, Ioannidis et al. (2023) used WRF-Chem version 3.9.1.1 and this paper used WRF-Chem version 4.3.3. Due to the major version change, the Fuentes et al. (2010) parameterization is no longer compatible with our sea-spray emission scheme and would have to be re-implemented from scratch in order to be tested.

12. Line 147: Mention what is HYCOM-CICE and TOPAZ

The proposed modification has been made.

13. Line 156: Define the abbreviation "NEMO"

The proposed modification has been made.

14. Line 157: Define the abbreviation "NANUK" and move the definition of neXtSIM in the previous line.

The proposed modification has been made.

15. Line 168: Define the abbreviation "MODIS".

The proposed modification has been made.

16. Line 194: Define the abbreviation "ERA5"

> The proposed modification has been made.

17. Line 195: Define the abbreviation "MERRA-2".

> The proposed modification has been made.

18. Line 197: Use Na+ for sodium and define once first appearing in the text.

> The proposed modification has been made.

19. Line 215 and Figure 2: You say you are focusing on the high Arctic. However, have you considered to use the radar data for sea ice at Utqiaġvik, Alaska to address the uncertainties in the different products you are using near the coastlines and even correct your parametrization for coastline regions. May et al. 2016 and Kirpes et al. 2019 are discussing radar data so they might be available.

> The objective of this work is not to validate and/or correct existing sea ice lead data sets, but rather to show what can be obtained when using what is currently available, hence why we use 2 models and 1 observational lead dataset, which are conceptually different. We acknowledge that these products could be compared with sea ice radar data, but this is out of the scope of the present paper.

20. Figure 3 and Figure 4: Show results for Salter et al. 2015 not only Gong et al. 2003. Or at least justify why you are showing results only for Gong et al. 2003. In Figures 5, 6 and 7 you are showing results for both so it's really confusing.

> Figures 3 and 4 now include the average of the three source functions instead of only Gong 2003 (see response to comment #2).

21. Figure 7: Replace sodium with $Na^+$. Also mention if observations are sub-micron, supermicron, PM2.5 or PM10.

> The Figure title has been adjusted according to the suggestion, and details about the cut-off size of observations are now given.

22. Lines 269-272: Add a reference.

> References to the corresponding Figure have been added.

23. Lines 290-291: Provide more information about the observations, are they sub-micron? Super-micron? Unclear.

> See response to comment #21.

24. Lines 307-310: Compare/discuss why you didn't choose Fuentes et al. 2010 or Ioannidis et al. 2023.

> See responses to comments #10 and #11.

25. Lines 325-327: Please re-read Kirpes et al. 2019. They did not measure atmospheric concentrations. They measured elemental fractions. Also be more precise. They used ratios as an indicator of local influence or transported and they found a local influence on sea spray aerosols. Re-write this part.

> The initial wording was poorly chosen, we did not mean that they measured concentrations but that measurements came from sea spray suspended in the air.
>
> This part has been re-written from:
>
> "*One possible explanation for this discrepancy is the transport of sea spray. Since Kirpes et al. (2019) measure atmospheric concentrations, while we only look at the OF at emission, their data can include sea salt transported from open ocean, which according to Figure 8 would show a higher OF at that season.*"
>
> To read:
>
> "*One possible explanation for the discrepancy between the OF we model and the measurements by Kirpes et al. (2019) is that their observations may include some transported / aged sea spray, and not only freshly emitted aerosols. Figure 8 shows that we predict a higher OF for open ocean sea spray than for leads during the season of the measurements. Therefore, if the conceptual model considered not only leads but also transported open ocean sea spray, we would obtain a higher OF, closer to the measurements. Furthermore, uncertainties also arise from the chlorophyll-a product as there is not enough observational data and there are model biases in both under-ice phytoplankton phenology and sea ice coverage (Wakamatsu et al., 2022). Additionally, massive phytoplankton blooms have been shown to occur under sea ice (Arrigo et al., 2012), which are likely missed by the model since it does not account for light availability*

*under sea ice. Therefore, the use of modeled chlorophyll-a data has significant uncertainties, even more so for its use in leads."*

26. Line 332 and Figure A1: Again, you only show results based on Gong et al. 2003. Please decide and show only Gong et al. 2003, and justify why, or both Gong et al. 2023 and Salter et al. 2015.

We now show the average of the three source functions instead of only one.

27. Figure 9: Compare with Fuentes et al. 2010.

See responses to comments #10 and #11.

28. Line 376: You are using WRF-Chem model which uses a different source function to estimate sea spray aerosols compared to your analysis so far. Why did you not include Ioannidis et al. 2023 source function in your analysis so far? You could add the source function in Figure 2 and in the text, as the other two (see earlier comment). It would make much more sense to include the source function you are going to use in your WRF-Chem simulations in your earlier analysis and apply there the Nilsson et al. 2001 parametrization/ratio you got and trying different sea ice products. Maybe you will get different results. Or if you already tested it then please discuss it/mention it in the text. Would be interesting to see how Ioannidis et al. 2023 compares with the rest in your analysis, since it was tested for simulations over the Arctic.

The Ioannidis et al. (2023) source function is now included all throughout the analysis, including in the conceptual model. All figures and associated text have been updated accordingly.

29. Define what SSA is in the WRF-Chem model version you are using.

The following is now included at line 378:

*"Sea spray emissions in WRF-Chem consist of sodium, chloride, sulfate and organics. The sulfate fraction is computed based on the measurements from Calhoun et al. (1991), and the Vignati et al. (2010) organic fraction is used for primary marine organic emissions (which are attributed to the organic carbon (OC) WRF-Chem species in this model version)."*

30. Line 380: You are using NCEP FNL reanalysis data. I assume sea ice fraction is coming from there. Did you consider using NEMO-netXtSIM sea ice concentration in your

WRFChem simulations, which gave you the best lead estimations? Can you use NEMO-netXtSIM in your simulation instead of NCEP FNL sea ice fraction? Or did you compare NCEP FNL from the three sea ice products you used to estimate potential uncertainties you might get in your model results coming from NCEP FNL?

Like lead fraction, sea ice fraction in the WRF-Chem simulations is taken from NEMO-neXtSIM. This is now made clearer with the following sentence at line 378:

*"[...] sea ice concentration is greater than 80%. Sea ice concentration is also taken from neXtSIM in all simulations."*

By default, irrespective of the meteorological forcing we use for polar simulations with WRF-Chem, we replace sea ice concentration provided by NCEP FNL (or ERA5) with the TOPAZ product studied in the conceptual model in this manuscript. Therefore, the difference between using the default WRF-Chem sea ice versus the neXtSIM sea ice is already tested with the conceptual model, and we do not test this sensitivity again with WRF-Chem.

31. Also, for your simulations are using NCEP FNL while for your earlier estimations you used ERA5. ERA5 and NCEP FNL have some differences. Try to explain the inconsistencies and the uncertainties you introduce to your parametrization by switching reanalysis data.

We have shown that FNL produces more reasonable meteorological fields for the Arctic in our past work, when combined with WRF-Chem at 100 km resolution (Marelle et al., 2021). However, for the conceptual model we use sea ice data at higher resolution (~12km), so we need a higher resolution product such as ERA5. Therefore, we use FNL for the online predictions with WRF-Chem, and the high resolution ERA5 data for the conceptual model. Furthermore, the conceptual model and the WRF-Chem parts of this work are not meant to be directly compared as one looks at emission fluxes only in an idealized framework and the other one is focused on aerosol concentrations in a 3D model with chemistry. Finally, WRF-Chem is only nudged to FNL at large scales but simulates its own meteorology. Therefore, even if WRF-Chem was driven by ERA5, there would be large differences between WRF and ERA5 anyway.

32. Line 370 and after: You add a new source in the model. How does this affect dry deposition in the model? How does this influence other inorganic aerosols?

Answering this question would require both additional sensitivity runs and re-completing our simulations. While these are two interesting questions, they do not warrant the computing time necessary, as they are out of the scope of the present study. These questions should be addressed in follow-on studies with WRF-Chem or global models that pick up this parameterization for use.

33. Line 380: You just mention the name of the model options for your simulations in the APPENDIX. Add text where you explain briefly the most important options for your simulations (such as aerosol scheme). Add citations that WRF-Chem has been used for Arctic simulations and that can be trusted.

The following sentence is now included at line 378 to expand on important model options:

*"Sea spray emissions in WRF-Chem consist of sodium, chloride, sulfate and organics. The sulfate fraction is computed based on the measurements from Calhoun et al. (1991), and the Vignati et al. (2010) organic fraction is used to primary marine organic emissions (which are attributed to the organic carbon (OC) WRF-Chem species in this model version). Blowing snow emissions of sea salt aerosol are also activated, following the implementation from Marelle et al. (2021). The aerosol scheme is the Model for Simulating Aerosol Interactions and Chemistry (MOSAIC) with 4 bins (Zaveri et al., 2008) and the initial and boundary conditions for atmospheric composition are taken from the Community Earth System Model 2.2 with the Community Atmosphere Model with Chemistry (CESM2.2: CAM-Chem) (Tilmes et al., 2022)."*

The following sentence has been included at line 366 to give context on the reliability of WRF-Chem in polar regions based on examples from the literature:

*"The WRF-Chem model is commonly used for Arctic case studies and generally shows good performance in reproducing atmospheric composition and aerosols in this region, including sea spray (Marelle et al., 2017, 2021; Raut et al., 2022; Ahmed et al., 2023; Ioannidis et al., 2023)."*

34. Line 418: Did you test how different will be your results if you use Fuentes et al. 2010 or Ioannidis et al. 2023 with adjusted Fuentes?

As indicated in the responses to comments #10 and #11, we have good reasons for not computing organic fractions with the Fuentes et al. (2010) / Ioannidis et al. (2023) parameterization.

35. Figure 10: You present your parametrisation implemented in a regional transport model. How do you make sure that your parametrization is trusted? How does the model compare against observations?

This is addressed in our response to comment #36, where we compare the model with observations. As with all model parameterizations, further testing and refinement will be needed before this can be fully trusted for use for all conditions / seasons.

36. Please compare against observations. How do you know your results make sense and can be trusted? Compare against sub-micron, super-micron and PM10 Na$^+$ observations at least in a few Arctic sites. Also include comparison against Cl$^-$.

> As per the reviewer's request, Figure A5 is now included in the Appendix, which compares surface concentrations of Na$^+$, Cl$^-$ and OC between WRF-Chem and observations at Zeppelin observatory for the simulated period, and the following text is now included at line 388:
>
> *"The WRF-Chem simulations are evaluated against measurements (Na$^+$, Cl$^-$ and OC aerosols) from the Zeppelin observatory (Ny-Ålesund, Svalbard), accessed from EBAS (Norwegian Institute for Air Research, 2024). This evaluation is presented in Figure A5 and shows a good performance of the model for all three species considered. Figure 7 also shows that the magnitude of monthly averaged Na+ concentration from the WRF-Chem simulations compare well with observations at Alert and Utqiagvik."*
>
> The simulation domain is deliberately small in order to limit the inclusion of open ocean, as numerical noise in the meteorology could induce important changes in open ocean sea salt emissions between the two simulations, which we want to avoid in order not to hamper the estimation of the impact of leads on sea salt concentration. As a result of the simulation domain being small, it does not comprise the location of many Arctic stations, hence the restriction to a comparison to only Zeppelin station.

37. Did you estimate ratios, such as Cl$^-$:Na$^+$, SO$_4^{2-}$:Na$^+$ as Kirpes et al. 2019 or did you calculate depletion factors as Frey et al. 2020 to be able to provide more robust conclusions about your model results?

> The ratios provided in Kirpes et al. (2019) are for wintertime, and the Frey et al. (2020) data was measured in Antarctica. Therefore neither of these measurements can be directly used to compare to our summertime WRF-Chem simulation. However, it would be interesting in future work to look into the results of the parameterization in new regions and time periods, in order to evaluate against detailed measurements such as Kirpes et al. (2019) or Frey et al. (2020).

**Reviewer # 2**

This study makes use of measurements of particle emission flux from Arctic leads to produce a parameterisation of the process that is suitable for models. The work is a novel contribution to a research question that is highly relevant to the scope of ACP, since this is a key polar aerosol process that is missing in current state-of-the-art climate models. The parameterisation is tested both using inputs from reanalysis to calculate emissions, and within the Weather Research and Forecasting model with chemistry (WRF-Chem). Results are compared to measurements where possible, and the relative importance of leads as an aerosol source compared to blowing snow and transported sea salt is assessed. I recommend that this article be published following some minor comments detailed below.

*General comments:*

1. Emissions from the marginal ice zone (MIZ) are largely excluded from the parameterisation. I acknowledge the justification for doing this in section 2.2.4, and analysis of the sensitivity of the results to this choice in Figure A1 and section 3.6. However, given the importance of the MIZ in the question of how Arctic aerosol will respond to future warming, some more discussion of modelling emissions from the MIZ would be welcome in the introduction and in 3.6 or 4. Specific recommendations about this are included below.

> We acknowledge that the question of the MIZ was eluded in the manuscript as not much is known in terms of aerosol emissions in this region (hence the high Arctic mask throughout this work). In practice, in 3D models like WRF-Chem, a 1-sea_ice mask is applied everywhere, including in the MIZ, to weight open ocean emissions. Once the community has a better understanding of sea spray emission processes in the MIZ, future work can improve on this. More discussion on this is now included as per the reviewer's suggestions (see comments #5 and #14).

2. Regarding figure 7 and the associated discussion, the potential role of transport and wet removal has been neglected as a potential source of disagreement between the conceptual modelling and the measured concentrations. Please address this in the text (see below).

> We acknowledge that ignoring transport and deposition in Figure 7 is an important limitation, although for the purpose of this section the method still provides useful information. This part is now better discussed as per the reviewer's suggestion (see response to comment #13).

3. There is some confusion caused by terminology in figure labels and captions, for example mixing "monthly sum" and "daily average" in the same figure or using the word "flux" both for emissions per unit area per unit time and simply for emissions per unit time. Please address these so that there is consistency throughout.

The labels and captions have been clarified and made consistent throughout as per the reviewer's comment (see responses to comments #10–12).

*Specific comments:*

4. Lines 21-22: sea spray has also been shown to have a key effect on the climate impact of secondary aerosol particles in the changing Arctic climate system, by changing the condensation sink (Browse et al. 2014) but this is by no means well established because results vary across models (Gilgen et al. 2018).

The following is now included after line 22:

*"Similarly, changes in sea spray aerosols can impact the condensation sink in the Arctic, which in turn affects new particle formation and therefore the CCN populations (Browse et al., 2014), although results vary for different models (Gilgen et al., 2018)."*

5. Line 33-35: it is important to highlight (here or elsewhere in introduction) why sources from leads are an important factor in establishing the sensitivity of Arctic aerosol to sea ice in the present day and in the future (or other differing climate states). Eg in the warming Arctic as sea ice extent/thickness changes and the marginal ice zone becomes wider at certain times of year (Strong and Rigor 2013), emissions from leads may be relatively more or less important than emissions from open ocean or pack ice. So without including this process in models, our knowledge of the changing natural aerosol baseline in the Arctic may be incomplete. (This point is made via the inclusion of the question in line 76, but the motivation could be made stronger).

This is a good point. The following has been added at line 39:

*"In particular, in the warming Arctic as sea ice extent/thickness changes and the marginal ice zone (MIZ) becomes wider at certain times of year (Strong and Rigor, 2013), emissions from leads may be relatively more or less important than emissions from the open ocean or pack ice. Not including this process in models may lead to an incomplete knowledge of the changing natural aerosol baseline in the Arctic, which may affect our representation of the Arctic climate."*

6. Figure 1b: since the Salter et al emission parameterisation includes an SST dependence, it's not clear whether the size distribution shown here in the figure is for a particular value of SST. Please give this information in the figure or caption.

The information on the SST used for the Salter source function is now included in the figure caption.

7. Line 128: "…only affect the sea salt emissions, but not…" surely it will also affect the organic emissions? Perhaps clearer to say "only affect the composition of the emitted particles, but not the absolute emissions flux of sea spray".

> Accordingly with the suggestion, line 128 now reads:
> *"[...] only affect the partitioning between inorganic sea salt and marine organics, but not the total sea spray emission flux."*

8. Line 195: the important conclusion that emissions from leads/blowing snow could be as important for the Arctic aerosol budget as transport relies on the comparison with MERRA-2 data. As such, please give some information here on how MERRA-2 sea salt aerosol is modelled (emission, removal processes) - has the representation of aerosols in MERRA-2 been previously evaluated for the Arctic? In particular I think it is important to highlight any uncertainties in the calculation of sea salt transport in MERRA-2 that could impact the comparison.

> MERRA-2 is indeed quite uncertain, especially in polar regions. The following discussion has been added at line 197 to give some context regarding the uncertainties in MERRA-2:
>
> *"Emissions of sea salt in MERRA-2 are computed online at hourly resolution (Randles et al., 2017). Although no validation of MERRA-2 sea salt aerosol is carried out in this work, previous studies on polar aerosols have relied on MERRA-2 data and showed reasonable performance of this product at high latitudes (Xian et al., 2022; Zamora et al., 2022; Böö et al., 2023). However, Lapere et al. (2023) found a large positive bias in sea salt aerosol surface concentrations in MERRA-2 compared to Arctic stations. Therefore, the sea salt aerosol transport computed here is probably an overestimation."*

9. Lines 202-203 and figure 2: it's not clear which underlying emissions flux is used to calculate the fluxes presented in figure 2(d-f) (although the figure caption for figure 3 states Gong (2003) is used, which suggests the same is true for figure 2). Please give this information here in the text or in the caption for figure 2, or both.

> The caption in Figure 2 now includes details on the source function used. This information is now also included at line 215.

10. Figure 3 and 5: the y-axis label gives these values as an "average rate" in #/s, while the figure caption states "monthly sum of sea salt number emissions". As such, I struggle to interpret whether the values represent total number of emitted particles calculated over each month (with units of particle number), or an average emission rate of particles for the month. Moreover, the y-axis label refers to a "daily average" which conflicts with the monthly aggregation here. I interpret this to mean that the aggregation is sampled from daily mean flux values, but I don't think it's necessary to write that here. The daily

meaning is described in the text (lines 120, 163, 195) and I think that's sufficient. Alternatively if I have interpreted this wrong, please clarify the meaning of "daily average" as used here.

> These daily averages were indeed confusing. All these are now converted to monthly sum and the corresponding figures and text have been modified accordingly.

11. Figure 4: similar to previous comment on figures 3 and 5, same note here about the use of the phrase "daily average" to describe seasonal means.

> Fixed.

12. Figures 7 and 9: values in maps are given units of [mass per unit time], whereas I would expect [mass per unit area per unit time] for a mass flux, and this would seem to be more consistent with your description of normalising the flux values by PBLH (in fig 7). Please double check these units.

> The units are fixed accordingly.

13. Figure 7 and line 294-296: comparing emissions data to concentration measurements: "…which signals that our parameterization could be improved despite already showing better agreement…" The conceptual model is of local emissions only (with some accounting for vertical mixing via the use of PBLH) whereas measured concentrations have also been influenced by removal and transport. As such this conceptual model of the "usual approach" is not totally consistent with what a climate model would produce when including explicit representations of the vertical mixing, transport, and removal processes. Given the fluctuations in transport and removal throughout the year at these locations, it seems possible that the wintertime underestimation and summertime overestimation by "usual approach" could be influenced by significant contributions from long-range transport and wet deposition (respectively). This should be emphasised in the text to highlight the uncertainties in modelling polar sea salt aerosol other than just these missing sources. Also, it may be informative to calculate the sodium concentrations for these locations from the WRF-Chem output (where transport and removal are included) and add those values here as datapoints for the months available.

> We acknowledge that this conceptual approach with a normalization of emissions is not ideal for comparison with observations. However, by adopting this simple method, the annual cycle at Alert and Utqiagvik using the "usual approach" is very similar to the annual cycle of concentrations simulated by CMIP6 models at these two stations (as shown in Lapere et al. (2023)). Therefore, this simple method, although not directly comparable to observations, as noted by the reviewer, due to the absence of transport and removal, is still able to account for the seasonal variations produced by climate models. As a result,

we use the same approach with leads and blowing snow to assess whether these sources modify this idealized annual cycle. This information was missing in the previous version of the manuscript and is now included line 303:

*"Although the method used here to compute the proxy for concentrations has major limitations as it does not account for transport nor removal (wet and dry), the annual cycle obtained with the* usual approach *is similar to the annual cycle of concentrations yielded by climate models from CMIP6 (which use the* usual approach *for emissions) at these two locations, as shown in Lapere et al. (2023). Therefore, despite the important simplifications in Figure 7, it strongly suggests that local sea ice sources of sea salt aerosol need to be accounted for to obtain a decent annual cycle of Na+ concentrations in the coastal Arctic."*

The WRF-Chem monthly averages at the 2 stations are now included in Figure 7 as per the reviewer's suggestion.

14. Figure A1: it would be instructive to see maps of the extent of the revised "high Arctic" areas using the different sea ice thresholds, to understand how the choice of threshold influences the spatial area under consideration. Moreover, one interesting application of the parameterisation introduced in this study will be to study the effect of sea ice changes on the future Arctic aerosol budget – specifically, the changing extent of the MIZ relative to total ice extent and how this may change the relative importance of open ocean/transported sea spray particles vs particles from open water areas in sea ice regions. It would be good to add some discussion of this aspect of the research either here or in the conclusion section. Specifically, it is important to emphasise the need for a better understanding of aerosol sources in the MIZ (including sea spray) and a need for this to be translated into climate models via parameterisations such as the one presented in this study.

In Figure A1, the extent of the high Arctic is kept the same as defined in the Methods section, irrespective of the lead threshold, in order to compare the total fluxes for the same area and only extract the sensitivity to the definition of a lead. An additional panel in Figure A1 is included to show the change in lead fraction resulting from changing the threshold, and the high Arctic mask is also shown on this figure to make it clear that the area where fluxes are computed does not change with the threshold.

Regarding the second part of the comment, the following paragraph is now included in the Conclusions, after line 449:
*"Finally, the parameterization proposed in this work can be leveraged to study the relative importance of open ocean/transported sea spray particles versus particles from open water areas in sea ice regions, in the context of changes in sea ice and extent of the MIZ. However, as highlighted here, a better understanding of aerosol sources in the MIZ and associated parameterizations are still missing to better comprehend Arctic aerosols."*

15. Line 444-445: it's shown in Figure 1 that it is not just the magnitude of the emissions that is uncertain, but even the sign of the response with increasing wind speed is unconstrained. This is an interesting result that could be stressed here.

> The following has been added at line 445:
> *"In particular, not only the magnitude but also the sign of the variation of emissions from leads with wind speed is not clearly established, depending on the chosen $R_{Nilsson}$, as illustrated in Figure 1a, with increasing $R_{Nilsson\ max}$ but decreasing $R_{Nilsson\ min}$ as a function of wind speed."*

*Technical corrections:*

16. Line 92: "the total particle number aerosol flux over the open Arctic Ocean was…" open ocean measurements reported in Nilsson are from the Barents Sea region further south rather than the Arctic Ocean itself. Please double check this and amend your statement here.

> The formulation was indeed too vague. This statement now reads:
> *"They found that the total particle number aerosol flux over the open Norwegian Sea and Barents Sea was described by […]"*

17. Figure 6: y-label is not very intuitive, suggest to change to something like "Total sea salt source / Gg month-1"

> The y-label has been modified accordingly.

18. Figure 8: please explain exactly which areas are included in "open ocean", i.e. what's the southerly extent.

> This was indeed unclear. To clarify, we are now showing the whole open ocean domain in the map panels and add to the caption the clarification that all areas with no sea ice inside this domain are considered in the open ocean calculation.

19. Figure 9: the map of Utqiagvik here uses a different projection from that in figure 7 - suggest to use a consistent projection for both.

> This is now fixed and both figures use the same projection.

20. Line 427: "This difference mass and number" should read "difference in"?

This is now adjusted.

21. Lines 638-end: should the references by Ólason et al. come under "O"?

This is now adjusted.

**Reviewer #3**

Lapere et al present an important modeling study of the contribution of leads to sea spray aerosol in the high Arctic. This work is particularly important in the context of rapidly declining sea ice in the Arctic. The manuscript is well-written. A main concern is that it seems from the methods that areas with sea ice concentration less than 80% are not considered for sea spray aerosol production in this study (according to my understanding of the statement on lines 178-179). While I understand this threshold for defining sea spray aerosol from leads, it is not clear how sea spray aerosol from sea ice regions with less than 80% sea ice concentrations is considered (according to my understanding of the statement on lines 178-179, it appears to be ignored/not considered). This needs to be clarified throughout, especially in the context of comparing lead-based sea spray aerosol and blowing snow aerosols, as sea spray aerosol emissions from within the Arctic, including the MIZ regions, are expected to increase the contribution of fresh sea spray aerosol to the high Arctic. In addition, as discussed below, the authors choice of sea ice lead satellite product also impacts the sea spray aerosol produced from leads, and this needs to be clear in the abstract and conclusions in particular. Given the lack of consideration of MIZ and choice of sea ice lead product (as shown in Figure 3), this means that the modeled sea spray aerosols are likely lower bounds. This is important as Arctic sea ice is declining and thinning, making sea spray aerosol emissions expected to be increasing across the Arctic, especially in the lower Arctic latitudes. Since the authors seem to focus on the High Arctic (e.g. Fig 2), it is then confusing when the authors then compare to coastal locations (e.g. Fig. 7), while not including sea spray aerosol from regions of sea ice conc < 80% (which it seems the authors had defined as such in Fig 2?). These points need to be considered and clarified in this modeling study. Additional detailed comments are provided below.
* * *
As understood by the reviewer, in the conceptual model we do not consider sea spray emissions (leads, blowing snow, or other) outside areas where sea ice concentration is greater than 80%, as emission processes in the marginal ice zone are quite uncertain. Essentially, we have no estimate of what the emission flux would be in the MIZ. This is now clarified where needed.

We agree that the values we provide are likely lower bounds, consistent with the reviewer's arguments. This is now mentioned in the conclusions.

The modifications needed related to these general comments are detailed in the responses to the specific comments.
* * *
1. Abstract: It is important that the authors clarify here that "sea ice regions" here pertains only to regions with >80% sea ice concentration, according to Section 2.2.1. It is also not clear here how "open ocean" is defined – is it <80% sea ice concentration? The definition of "high Arctic" in Figure 2 is different than in many studies and needs a statement in the abstract for clarification. Overall, these points need clarification in the abstract and elsewhere, for the reasons noted above.

> Line 7 now reads: *"[...] for 0.3%-9.8% of the annual sea salt aerosol number emissions in the Arctic Ocean regions where sea ice concentration is greater than 80%."*
>
> The notion of open ocean really only applies in Section 3.7 (WRF-Chem) since throughout the conceptual model part, only areas with >80% sea ice concentration are considered. In Section 3.7, we define the MIZ as sea ice concentration <80% and >15%. Below 15% we consider that there is open ocean. This is now made clearer line 387: *"[...] no sea spray is considered in the marginal ice zone, between 15% and 80% sea ice concentration, and open ocean is considered where sea ice concentration is below 15%."*

2.  Section 2.1.2 and Figure 8 discussion: This section parameterizes the organic fraction of marine aerosols based on chlorophyll-a. However, Quinn et al. (2014, Nat. Geosc.) found through observations that the organic content of sea spray aerosol does not vary with chlorophyll-a levels. Rocchi et al. (2024, Environ. Sci. Technol.) also found no correlation with chlorophyll for Arctic sea spray aerosol. Both Kirpes et al. (2019, ACS Central Sci.) and Zeppenfeld et al. (2021, ACS Earth Space Chem) found increased seawater and sea spray aerosol organic content associated with the presence of sea ice. Given these findings, it is critical that the authors add discussion of the uncertainties and caveats associated with parameterizing the organic fraction as a function of chlorophyll.

> We agree and have added some sentences in Section 2.1.2 to address this:
>
> *"We acknowledge that there are limitations associated with parameterizing organic fraction using chlorophyll-a. However the availability of chlorophyll-a products makes it more likely that these parameterizations can be adapted and used within climate and Earth system models. While it is clear that other seawater characteristics than chlorophyll-a (such as organic carbon or glucose concentration) can better account for the OF of sea spray (Fuentes et al., 2010; Quinn et al., 2014; Rocchi et al., 2024), their lack of general availability in satellite or model-based products limits our ability to recommend these now for use by large scale models."*

3.  Lines 128-130: The authors assume that the size distribution of marine organics is the same as inorganic sea salt, but this is not in line with sea spray aerosol studies (Prather et al., 2013, PNAS), including in the Arctic (Kirpes et al., 2019, ACS Central Sci.). Increased marine organics are present in smaller particles produced from film drops (Wang et al. 2017, PNAS).

> This comment is important and points out the gap between what is known from specific observational studies and what can be generalized for use in models. We are not aware of any available data providing detailed size distributions of marine organic emissions, especially from leads, that can be implemented in models. In addition, the size distribution of sea spray from leads is itself uncertain. Therefore, we are not able to assume more or

less organics than open ocean, despite the findings from Prathers et al. (2013), Wang et al. (2017) and Kirpes et al. (2019).

The following has been included at line 130:
*"Future work that refines the size distribution of marine organic emissions from leads is needed, as measurements have shown that inorganic sea salt and marine organics have different dominant modes (Prather et al., 2013; Wang et al., 2017), including in the Arctic (Kirpes et al., 2019)."*

4. Sections 2.1.3 and 3.2: Prior sections generally do a good job discussing uncertainties associated with uncertainties in parameterizing sea spray aerosol from leads and the need for more measurements. However, similar discussion of the uncertainties associated with parameterizing blowing snow sea salt aerosol emissions is missing here. As written it comes across that sea salt aerosol fluxes from leads are more uncertain than blowing snow, when in actuality sea spray aerosol from leads have been studied for decades, whereas Arctic blowing snow parameterizations are missing observational evaluation and flux measurements. Therefore, the authors should add discussion of the blowing snow sea salt aerosol uncertainties just as they have for sea spray aerosol. This includes discussion of the uncertainties associated with their assumptions and the need for measurements to evaluate these.

The blowing snow parameterization used here comprises thresholds on surface temperature and wind speed to determine whether blowing snow events are possible or not. We acknowledge that these thresholds were calibrated/validated using measurements in wintertime in Antarctica (see Yang et al., 2019) and may therefore not work as well for summertime Arctic conditions. However, the same parameterization was used and validated for the Arctic, including in spring and summer months, in Gong et al. (2023) and Confer et al. (2023), from which we used the recommended tuning parameters. Therefore, we rely on the validations made in Gong et al. (2023) and Confer et al. (2023). It is worth noting that we obtain annual total emissions of sea salt from blowing snow comparable to Confer et al. (2023).

In order to discuss uncertainties in the blowing snow fluxes we compute and explain that we do not investigate them as the focus is on leads, the following is now included at line 136:

*"Although this blowing snow parameterization bears uncertainties, including snow salinity, number of sea salt aerosols released per snow grain, and fitting parameters (threshold wind speeds, surface temperatures, effect of snow age…), this work does not aim to investigate these uncertainties. The tuning parameters used here correspond to the ones used in Gong et al. (2023) and Confer et al. (2023) where more extensive validations against Arctic observations were conducted. In particular, Confer et al. (2023) showed that using variable or fixed snow salinity can change the total emission flux by 72%. As a result, the blowing snow fluxes are given as indicative for comparison with leads, but are*

> *not representative of the range of possible actual values, which are still uncertain and would require more observations to be estimated.”*
>
> We also include a comparison with the results from Confer et al. (2023) at line 266:
>
> *“For blowing snow, we find a total sea salt aerosol emission flux of 0.2-0.3 Tg yr⁻¹ for this region, which is consistent with the results of Confer et al. (2023) who found 0.28-2.24 Tg yr⁻¹ for a larger region of the Arctic.”*

5. Section 3.1: The authors state that "we cannot consider ArcLeads as viable surface information", but a proper, thorough sea ice evaluation study is not conducted to support this statement and others in this section. Is there a paper that compare the accuracy of ArcLeads vs TOPAZ and neXtSIM? My understanding is that satellite sea ice product evaluation is outside of the expertise of the authors, and therefore it is inappropriate to discuss the accuracy of the sea ice products without referencing other studies dedicated to this purpose. It is clear from Figure 2 that the choice of sea ice lead product will have a major impact on the sea spray aerosol emissions so this discussion is critical to the results presented. Therefore, the authors need to add additional sea ice literature and further discuss caveats/assumptions.

> The reason why we discard ArcLeads is not related to the values it provides or to an evaluation of this product, but rather because what we need in this work is not so much a lead detection product (which can include leads covered by sea ice, by definition) as an "open lead" product. The following discussion is now included at line 214 to clarify this:
>
> *"This explains why the lead fraction from ArcLeads is so much larger than the lead fractions we derive from TOPAZ and neXtSIM. Leads by definition can be covered by thin ice (WMO, 2014) and that is what ArcLeads provides. But for our study we only need open water leads. Thus, this means that using ArcLeads provides an overestimation of the upper bound of sea spray fluxes from open leads, probably by one order of magnitude, and so ArcLeads is not usable as is for our sea spray aerosol parameterization. Lead fraction can vary by one order of magnitude in observational lead products depending on the considered physical lead properties, as reported by von Albedyll et al. (2024). There are satellite products that can provide open water fraction based on ice divergence (e.g. von Albedyll et al., 2024), but they are currently not available on the spatio-temporal scales needed for our study."*

6. Line 240: This sentence states that blowing snow occurs in the summer. What is the justification for this at the time when melt ponds are present? What evaluation was completed to evaluate the presence of blowing snow and support this sentence?

The wording at line 240 may have been misleading, as the blowing snow fluxes we obtain in summer are, although not zero, very small compared to other months (colorbar values 10 times smaller in Figure 4 and log scale in Figure 6). This is now better reflected in the text.

In line with comment #4, we do not evaluate the blowing snow parameterization in this work, but rely on evaluations conducted in the literature and use the corresponding tuning parameters. In particular, Confer et al. (2023) include the summertime period in their work and obtain similar annual total emissions as we do.

Furthermore, the sensitivity test conducted in Figure A3 shows that the condition we impose on snow thickness variations (which partly translates to snow melt) helps dampen blowing snow emissions in summertime, so that the occurrence of melt ponds is somewhat accounted for here.

7. Lines 248-249, 257-258, & 427-428: To my knowledge, the size distribution of aerosols produced from blowing snow sublimation in the Arctic is highly uncertain and not yet observationally determined. This caveat needs to be discussed here, as the phrasing makes it sound like the blowing snow aerosol size distribution is known. Furthermore, Nilsson et al. (2001) and May et al. (2016) did not measure the aerosol size distribution from blowing snow, as implied.

We agree that the formulation was misleading. However, there is observational evidence that blowing snow events in the Arctic produce fine-mode sea salt aerosol (Gong et al., 2023), and that the size distribution in Antarctica is skewed towards smaller particles (Frey et al., 2020).

Line 248-249 now reads:
*"The difference between number and mass obtained here is connected to the blowing snow parameterization releasing mostly fine mode $Na^+$ aerosols in large quantities but significantly less coarse mode aerosol, and emissions from leads, in our parameterization, being much higher for particles bigger than 100 nm (Figure 1b)."*

Accordingly, line 257-258 now reads:
*"[...] is consistent with previous observations, which revealed mostly coarse size sea salt aerosol from leads (Nilsson et al., 2001; May et al., 2016) and mostly fine mode sea salt aerosols from blowing snow (Frey et al., 2020; Gong et al., 2023)."*

Line 427-428 now reads:
*"The asymmetry between number and mass is connected to the parameterizations of size distributions used in this work, which both for blowing snow and leads are still relatively uncertain and need further observational data."*

8. Lines 281-282 and 444: It is stated that the largest uncertainty is the uncertainty/scarcity of emission flux measurements, but it would seem, based on Figure 3, that the sea ice lead product used could result in even greater uncertainty. This should be discussed.

> The lead fraction difference in Figure 3 originates from a different definition of what a lead is. It is not a measurement or model uncertainty. We are interested in open water leads (i.e. open water fraction), which is not what the thermal infrared based satellite product provides (see response to comment #5). Using ArcLeads therefore provides an overestimation of the upper bound of sea spray from leads, which is why we do not consider it in our evaluation of uncertainties.
>
> To moderate this statement, lines 281-282 now read:
> *"As explained in Section 3.1, emissions from leads computed using ArcLeads are very different from the other two products because ArcLeads includes leads covered by thin ice. The differences between the fluxes obtained with ArcLeads and with the model-based products therefore originate from a different definition of what a lead is, rather than uncertainty in the models or the satellite product, and ArcLeads most likely provides an unrealistic upper bound of sea spray fluxes from leads."*

9. Section 3.4: According to Figure 2 and associated discussion, both Alert and Utqiagvik are outside of the High Arctic where leads are considered in this modeling study, and therefore, my understanding is that the MIZ mask would be relevant for these locations. But, according to Figure 7, it seems that local leads are considered, so this is confusing. Please clarify.

> We acknowledge this part needs clarification as a key assumption was missing. In general throughout this work we only consider leads in the high Arctic mask of Figure 2, as indicated by the reviewer, to avoid falsely computing leads in the MIZ. In the particular case of Section 3.4 we remove this condition but still consider leads only where sea ice concentration is greater than 80%.
>
> This now clarified after line 291:
> *"In this Section, we remove the condition of being inside the high Arctic mask of Figure 2 and consider that leads can be found wherever sea ice concentration is greater than 80%, including in coastal areas near the two stations considered. Although the validity of such an approach needs further investigation, it is necessary here as only coastal stations provide year-long observations with a robust annual cycle. Furthermore, grid points with sea ice concentration below 80% are ignored, since the objective of this part is to assess whether sea ice sources only can explain the seasonal variations of sea salt aerosols for these locations."*

10. Figure 7: The boxes around Alert and Utqiagvik in Figure 7 are very small – equivalent to maybe an hour or so of transport time (i.e. very little transport), when at least 24 h of transport time would be more appropriate. This is problematic for the comparison to local sodium aerosol, as a larger domain should be considered. What is the sensitivity of the

results shown to the box chosen? Also, how is open water (i.e. sea ice conc > 80%) considered for these model-observational comparisons? It seems like this is ignored? Please also add to the caption what size range is shown for the sodium surface conc.

> This is a good point. To circumvent this problem we have increased the size of the box. It is now 30 grid points from the station location in each direction, which corresponds to a 24h-transport time for a 4m/s mean wind speed. With this larger box, the figure remains quite similar to the previous version. As a test we also increased the size of the box up to 100 grid points (not shown in the manuscript), and obtained slight changes but a similar shape overall, showing the robustness of the results.
>
> Since we only consider emissions from leads and blowing snow in this part, open water is indeed ignored here, with the same criterion as before, i.e. all grid points with sea ice concentration below 80% are not considered. We acknowledge that in reality there is a mix of open ocean and sea ice emissions, especially in summer months, but the goal here is to show that sea ice sources only can, to some extent, explain most of the seasonal variations of sea salt aerosols for these locations. This is now specified in the manuscript after line 291.
>
> The text now indicates the aerosol sizes included in the observations.

*Additional Comments:*

11. Lines 61-63: Chen et al. (2022) did not prove resuspension of snowpack deposited sea spray aerosol, but could not rule it out because of leads directly upwind of the snowpack.

> This now reads: *"[...] and through deposition of sea spray onto snow which has the potential to be later re-suspended through blowing snow, although their measurements could not conclude on this last part."*

12. Lines 91: Held et al. (2011) also measured fluxes, but only at lower wind speeds.

> We intended to say that Nilsson et al. (2001) provided the only flux measurements that could be readily used in models, as Held et al. (2011) does not provide a direct formulation of the fluxes they observed as a function of wind speed or other meteorological parameter. Line 91 now reads:
> *"[...] detailed measurements of sea spray aerosol fluxes over leads that can readily be used for modeling purposes are found in Nilsson et al. (2001)."*

13. Line 176: Add the clarification that the "all areas of open water" is only in areas with sea ice >80%.

This is now re-written as:
*"[...] all areas of open water within the pack ice where the sea ice concentration is greater than 80%."*

14. Line 197: Please clarify what is meant by "transport from the lower latitudes". Does this include the MIZ and coastal Arctic?

Line 197 now reads: *"[...] transport from the lower latitudes (i.e. northward transport through the high Arctic contour in Figure 2)."*

15. Lines 209-210: It is important to note that freezing and opening of leads are processes that are faster than the timescale of the satellite observations.

The following is now included at line 209:
*"[...] are now covered by thin ice, as the freezing and opening of leads are processes that are faster than the timescale of the satellite observations."*

16. Lines 218-219: See also Rheinlænder et al 2024, JGR-Ocean, doi:10.1029/2023JC020395, "Breaking the Ice: Exploring the Changing Dynamics of Winter Breakup Events in the Beaufort Sea".

The following is now included at line 218:
*"[...] wind divergence in this area, along with thinner sea ice which leads to more frequent breakup events in winter (Rheinlænder et al., 2024)."*

17. Lines 220-221: Wouldn't the low spatial resolution of TOPAZ compared to typical lead sizes explain the lack of "discernible lead spaces".

The reviewer may be right, at least partially. It would be interesting to run a higher resolution simulation of TOPAZ and verify this. Note that it has been reported in previous literature that sea ice deformation (and the subsequent lead opening) as simulated in sea ice models at horizontal resolution of about >5km is not well reproduced compared to observations when using, like in TOPAZ, a standard viscous-plastic rheology (see e.g. Bouchat et al. 2022). TOPAZ and neXtSIM are run with roughly the same horizontal resolution (12.5km for TOPAZ, 12km for neXtSIM) but neXtSIM exhibits lead patterns which do not show in TOPAZ, meaning that model resolution cannot, solely, explain the absence of lead-like shapes. One can argue that the use of a brittle rheology like in neXtSIM helps to simulate leads realistically, at least at relatively coarse spatial resolutions >5km.

18. Lines 266-267: Does "lower latitudes" here include within the Arctic? Please clarify.

> The words "lower latitudes" have been taken out for clarification, since the computation that is made here is all sea salt aerosol entering our high Arctic contour with a northward flux, whatever their origin (meaning the Arctic is indeed included). This sentence now reads: *"[...] mass of sea salt aerosol transported into the high Arctic of [...]"*

19. Line 274: This lacks acknowledgement of the uncertainty in blowing snow aerosol emissions.

> In line with comment #4, the purpose of this work is not to evaluate the existing blowing snow parameterization, but rather use it as a means of comparison. As a result, we do not explore the uncertainties associated with the blowing snow parameterization. However, this uncertainty is now acknowledged, following comment #7.

20. Line 326-327: Kirpes et al. (2019) did not measure concentrations and only considered nascent (locally produced) sea spray aerosol – not those transported from open ocean.

> The initial wording was poorly chosen, we did not mean that they measured concentrations but that measurements came from sea spray suspended in the air.
>
> This part has been re-written from:
>
> "*One possible explanation for this discrepancy is the transport of sea spray. Since Kirpes et al. (2019) measure atmospheric concentrations, while we only look at the OF at emission, their data can include sea salt transported from open ocean, which according to Figure 8 would show a higher OF at that season.*"
>
> To read:
>
> "*One possible explanation for the discrepancy between the OF we model and the measurements by Kirpes et al. (2019) is that their observations may include some transported / aged sea spray, and not only freshly emitted aerosols. Figure 8 shows that we predict a higher OF for open ocean sea spray than for leads during the season of the measurements. Therefore, if the conceptual model considered not only leads but also transported open ocean sea spray, we would obtain a higher OF, closer to the measurements. Furthermore, uncertainties also arise from the chlorophyll-a product as there is not enough observational data and there are model biases in both under-ice phytoplankton phenology and sea ice coverage (Wakamatsu et al., 2022). Additionally, massive phytoplankton blooms have been shown to occur under sea ice (Arrigo et al., 2012), which are likely missed by the model since it does not account for light availability under sea ice. Therefore, the use of modeled chlorophyll-a data has significant uncertainties, even more so for its use in leads.*"

21. Figure 9: Why is the Utqiagvik box bigger here than in Fig 7?

This is now corrected and made consistent with the same box in Figure 9 as in Figure 7.

22. Figure 10: The grey band is difficult to see.

This band is only designed to mask out the values at the edges of the simulation domain, therefore we have left it as is.

[revised manuscript text omitted]